# Federated Bayesian Optimization via Thompson Sampling

**Zhongxiang Dai**[†]**, Bryan Kian Hsiang Low**[†]**, Patrick Jaillet**[§]
Dept. of Computer Science, National University of Singapore, Republic of Singapore[†]
Dept. of Electrical Engineering and Computer Science, MIT, USA[§]
{daizhongxiang,lowkh}@comp.nus.edu.sg[†],jaillet@mit.edu[§]

## Abstract

*Bayesian optimization* (BO) is a prominent approach to optimizing expensive-to-evaluate black-box functions. The massive computational capability of edge devices such as mobile phones, coupled with privacy concerns, has led to a surging interest in *federated learning* (FL) which focuses on collaborative training of *deep neural networks* (DNNs) via *first-order optimization* techniques. However, some common machine learning tasks such as hyperparameter tuning of DNNs lack access to gradients and thus require *zeroth-order/black-box optimization*. This hints at the possibility of extending BO to the FL setting (FBO) for agents to collaborate in these black-box optimization tasks. This paper presents *federated Thompson sampling* (FTS) which overcomes a number of key challenges of FBO and FL in a principled way: We (a) use *random Fourier features* to approximate the Gaussian process surrogate model used in BO, which naturally produces the parameters to be exchanged between agents, (b) design FTS based on *Thompson sampling*, which significantly reduces the number of parameters to be exchanged, and (c) provide a theoretical convergence guarantee that is robust against heterogeneous agents, which is a major challenge in FL and FBO. We empirically demonstrate the effectiveness of FTS in terms of communication efficiency, computational efficiency, and practical performance.

## 1 Introduction

*Bayesian optimization* (BO) has recently become a prominent approach to optimizing expensive-to-evaluate black-box functions with no access to gradients, such as in hyperparameter tuning of *deep neural networks* (DNNs) [49]. A rapidly growing computational capability of edge devices such as mobile phones, as well as an increasing concern over data privacy, has given rise to the widely celebrated paradigm of *federated learning* (FL) [39] which is also known as *federated optimization* [35]. In FL, individual agents, without transmitting their raw data, attempt to leverage the contributions from the other agents to more effectively optimize the parameters of their *machine learning* (ML) model (e.g., DNNs) through *first-order optimization* techniques (e.g., stochastic gradient descent) [24, 33]. However, some common ML tasks such as hyperparameter tuning of DNNs lack access to gradients and thus require *zeroth-order/black-box optimization*, and a recent survey [24] has pointed out that hyperparameter optimization of DNNs in the FL setting is one of the promising research directions for FL. This opportunity, combined with the proven capability of BO to efficiently optimize expensive-to-evaluate black-box functions [49], naturally suggests the possibility of extending BO to the FL setting, which we refer to as *federated BO* (FBO).

The setting of our FBO is similar to that of FL, except that FBO uses zeroth-order optimization, in contrast to first-order optimization adopted by FL. In FBO, every agent uses BO to optimize a black-box function (e.g., hyperparameter optimization of a DNN) and attempts to improve the efficiency of

its BO task by incorporating the information from other agents. The information exchange between agents has to take place without directly transmitting the raw data of their BO tasks (i.e., history of input-output pairs). A motivating example is when a number of mobile phone users collaborate in optimizing the hyperparameters of their separate DNNs used for next-word prediction in a smart keyboard application, without sharing the raw data of their own hyperparameter optimization tasks. This application cannot be handled by FL due to the lack of gradient information and thus calls for FBO. Note that the generality of BO as a black-box optimization algorithm makes the applicability of FBO extend beyond hyperparameter tuning of DNNs on edge devices. For example, hospitals can be agents in an FL system [24]: When a hospital uses BO to select the patients to perform a medical test [61], FBO can be employed to help the hospital accelerate its BO task using the information from other hospitals without requiring their raw data. Unfortunately, despite its promising applications, FBO faces a number of major challenges, some of which are only present in FBO, while others plague the FL setting in general.

The first challenge, which arises only in FBO yet not FL, results from the requirement for retaining (hence not transmitting) the raw data. In standard FL, the transmitted information consists of the parameters of a DNN [39], which reduces the risk of privacy violation compared to passing the raw data. In BO, the information about a BO task is contained in the *surrogate model* which is used to model the objective function and hence guide the query selection (Section 2). However, unlike a DNN, a *Gaussian process* (GP) model [45], which is the most commonly used surrogate model in BO, is *nonparametric*. Therefore, a BO task has no parameters (except for the raw data of BO) that can represent the GP surrogate and thus be exchanged between agents, while the raw data of BO should be retained and never transmitted [29]. To overcome this challenge, we exploit *random Fourier features* (RFF) [43] to approximate a GP using a Bayesian linear regression model. This allows us to naturally derive parameters that contain the information about the approximate GP surrogate and thus can be communicated between agents without exchanging the raw data (Section 2). In fact, with RFF approximation, the parameters to be exchanged in FBO are equivalent to those of a linear model in standard FL (Section 3.2).

FBO also needs to handle some common challenges faced by FL in general: communication efficiency and heterogeneity of agents. Firstly, communication efficiency is an important factor in the FL setting since a large number of communicated parameters places a demanding requirement on the communication bandwidth [24] and is also more vulnerable to potential malicious privacy attacks [3]. To this end, we use *Thompson sampling* (TS) [54], which has been recognized as a highly effective practical method [4], to develop our FBO algorithm. The use of TS reduces the required number of parameters to be communicated while maintaining competitive performances (Section 3.2). Secondly, the heterogeneity of agents is an important practical consideration in FL since different agents can have highly disparate properties [33]. In FBO, heterogeneous agents represent those agents whose objective functions are significantly different from that of the *target agent* (i.e., the agent performing BO). For example, the optimal hyperparameters of the DNN for next-word prediction may vary significantly across agents as a result of the distinct typing habits of different mobile phone users. To address this challenge, we derive a theoretical convergence guarantee for our algorithm which is *robust against heterogeneous agents*. In particular, our algorithm achieves *no regret* asymptotically even when some or all other agents have highly different objective functions from the target agent.

This paper introduces the first algorithm for the FBO setting called *federated Thompson sampling* (FTS) which is both theoretically principled and practically effective. We provide a theoretical convergence guarantee for FTS that is robust against heterogeneous agents (Section 4). We demonstrate the empirical effectiveness of FTS in terms of communication efficiency, computational efficiency, and practical performance using a landmine detection experiment and two activity recognition experiments using Google glasses and mobile phone sensors (Section 5).

## 2 Background

**Bayesian Optimization (BO) and Gaussian Process (GP).** BO attempts to find a global maximizer of a black-box *objective function* $f$ defined on a domain $\mathcal{X} \subset \mathbb{R}^D$, i.e., find $x^* \triangleq \arg\max_{x \in \mathcal{X}} f(x)$ through sequential queries. That is, in iteration $t = 1, \ldots, T$, BO queries an input $x_t$ to observe a noisy output $y(x_t) \triangleq f(x_t) + \epsilon$ where $\epsilon$ is an additive Gaussian noise with variance $\sigma^2$: $\epsilon \sim \mathcal{N}(0, \sigma^2)$. The aim of a BO algorithm is to minimize *regret*. Specifically, if the *cumulative regret* $R_T \triangleq \sum_{t=1,\ldots,T} [f(x^*) - f(x_t)]$ grows sublinearly, then the BO algorithm is said to achieve *no*

*regret* since it implies that the *simple regret* $S_T \triangleq \min_{t=1,\ldots,T}[f(x^*) - f(x_t)] \leq R_T/T$ goes to $0$ asymptotically. In order to sequentially select the queries to minimize regret, BO usually uses a GP as a surrogate to model the objective function $f$. A GP is a stochastic process in which any finite subset of random variables follows a multivariate Gaussian distribution [45]. A GP, which is represented as $\mathcal{GP}(\mu(\cdot), k(\cdot, \cdot))$, is fully characterized by its mean function $\mu(x)$ and covariance/kernel function $k(x, x')$ for all $x, x' \in \mathcal{X}$. We assume w.l.o.g. that $\mu(x) \triangleq 0$ and $k(x, x') \leq 1$ for all $x, x' \in \mathcal{X}$. We consider the widely used *squared exponential* (SE) kernel here. Conditioned on a set of $t$ observations $D_t \triangleq \{(x_1, y(x_1)), \ldots, (x_t, y(x_t))\}$, the posterior mean and covariance of GP can be expressed as

$$\mu_t(x) \triangleq k_t(x)^\top (K_t + \sigma^2 I)^{-1} y_t , \quad \sigma_t^2(x, x') \triangleq k(x, x') - k_t(x)^\top (K_t + \sigma^2 I)^{-1} k_t(x') \quad (1)$$

where $K_t \triangleq [k(x_{t'}, x_{t''})]_{t',t''=1,\ldots,t}$, $k_t(x) \triangleq [k(x, x_{t'})]_{t'=1,\ldots,t}^\top$, and $y_t \triangleq [y(x_1), \ldots, y(x_t)]^\top$.

Unfortunately, GP suffers from poor scalability (i.e., by incurring $\mathcal{O}(t^3)$ time), thus calling for the need of approximation methods. Bochner's theorem states that any continuous stationary kernel $k$ (e.g., SE kernel) can be expressed as the Fourier integral of a spectral density $p(s)$ [45]. As a result, random samples can be drawn from $p(s)$ to construct the $M$-dimensional ($M \geq 1$) *random features* $\phi(x)$ for all $x \in \mathcal{X}$ (Appendix A) whose inner product can be used to approximate the kernel values: $k(x, x') \approx \phi(x)^\top \phi(x')$ for all $x, x' \in \mathcal{X}$ [43]. The approximation quality of such a *random Fourier features* (RFF) approximation method is theoretically guaranteed with high probability: $\sup_{x,x' \in \mathcal{X}} |k(x, x') - \phi(x)^\top \phi(x')| \leq \varepsilon$ where $\varepsilon \triangleq \mathcal{O}(M^{-1/2})$ [43]. Therefore, increasing the number $M$ of random features improves the approximation quality (i.e., smaller $\varepsilon$).

A GP with RFF approximation can be interpreted as a Bayesian linear regression model with $\phi(x)$ as the features: $\hat{f}(x) \triangleq \phi(x)^\top \omega$. With the prior of $\mathbb{P}(\omega) \triangleq \mathcal{N}(0, I)$ and given the set of observations $D_t$, the posterior belief of $\omega$ can be derived as

$$\mathbb{P}(\omega | \Phi(X_t), y_t) = \mathcal{N}(\nu_t, \sigma^2 \Sigma_t^{-1}) \quad (2)$$

where $\Phi(X_t) = [\phi(x_1), \ldots, \phi(x_t)]^\top$ is a $t \times M$-dimensional matrix and

$$\Sigma_t \triangleq \Phi(X_t)^\top \Phi(X_t) + \sigma^2 I , \qquad \nu_t \triangleq \Sigma_t^{-1} \Phi(X_t)^\top y_t \quad (3)$$

which contain $M^2$ and $M$ parameters, respectively. As a result, we can sample a function $\tilde{f}$ from the GP posterior/predictive belief with RFF approximation by firstly sampling $\tilde{\omega}$ from the posterior belief of $\omega$ (2) and then setting $\tilde{f}(x) = \phi(x)^\top \tilde{\omega}$ for all $x \in \mathcal{X}$. Moreover, $\Sigma_t$ and $\nu_t$ (3) fully define the GP posterior/predictive belief with RFF approximation at any input $x$, which is a Gaussian with the mean $\hat{\mu}_t(x) \triangleq \phi(x)^\top \nu_t$ and variance $\hat{\sigma}_t^2(x) \triangleq \sigma^2 \phi(x)^\top \Sigma_t^{-1} \phi(x)$ (Appendix B).

**Problem Setting of Federated Bayesian Optimization.** Assume that there are $N + 1$ agents in the system: $\mathcal{A}$ and $\mathcal{A}_1, \ldots, \mathcal{A}_N$. For ease of exposition, we focus on the perspective of $\mathcal{A}$ as the *target agent*, i.e., $\mathcal{A}$ attempts to use the information from agents $\mathcal{A}_1, \ldots, \mathcal{A}_N$ to accelerate its BO task. We denote $\mathcal{A}$'s objective function as $f$ and a sampled function from $\mathcal{A}$'s GP posterior belief (1) at iteration $t$ as $f_t$. We represent $\mathcal{A}_n$'s objective function as $g_n$ and a sampled function from $\mathcal{A}_n$'s GP posterior belief with RFF approximation as $\hat{g}_n$. We assume that all agents share the same set of random features $\phi(x)$ for all $x \in \mathcal{X}$, which is easily achievable since it is equivalent to sharing the first layer of a neural network in FL (Appendix A). For theoretical analysis, we assume that all objective functions are defined on the same domain $\mathcal{X} \subset \mathbb{R}^D$ which is assumed to be discrete for simplicity but our analysis can be easily extended to compact domain through discretization [9]. A smoothness assumption on these functions is required for theoretical analysis; so, we assume that they have bounded norm induced by the *reproducing kernel Hilbert space* (RKHS) associated with the kernel $k$: $\|f\|_k \leq B$ and $\|g_n\|_k \leq B$ for $n = 1, \ldots, N$. This further suggests that the absolute function values are upper-bounded: $|f(x)| \leq B$ and $|g_n(x)| \leq B$ for all $x \in \mathcal{X}$. We denote the maximum difference between $f$ and $g_n$ as $d_n \triangleq \max_{x \in \mathcal{X}} |f(x) - g_n(x)|$ which characterizes the similarity between $f$ and $g_n$. A smaller $d_n$ implies that $f$ and $g_n$ are more similar and heterogeneous agents are those with large $d_n$'s. Let $t_n$ denote the number of BO iterations that $\mathcal{A}_n$ has completed (i.e., number of observations of $\mathcal{A}_n$) when it passes information to $\mathcal{A}$; $t_n$'s are constants unless otherwise specified.

# 3 Federated Bayesian Optimization (FBO)

## 3.1 Federated Thompson Sampling (FTS)

Before agent $\mathcal{A}$ starts to run a new BO task, it can request for information from the other agents $\mathcal{A}_1, \ldots, \mathcal{A}_N$. Next, every agent $\mathcal{A}_n$ for $n = 1, \ldots, N$ uses its own history of observations, as well as the shared random features (Section 2), to calculate the posterior belief $\mathcal{N}(\nu_n, \sigma^2 \Sigma_n^{-1})$ (2) where $\nu_n$ and $\Sigma_n$ represent $\mathcal{A}_n$'s parameters of the RFF approximation (3). Then, $\mathcal{A}_n$ draws a sample from the posterior belief: $\omega_n \sim \mathcal{N}(\nu_n, \sigma^2 \Sigma_n^{-1})$ and passes the $M$-dimensional vector $\omega_n$ to the target agent $\mathcal{A}$ (possibly via a central server). After receiving the messages from other agents, $\mathcal{A}$ uses them to start the FTS algorithm (Algorithm 1). To begin with, $\mathcal{A}$ needs to define (a) a monotonically increasing sequence $[p_t]_{t \in \mathbb{Z}^+}$ s.t. $p_t \in (0, 1]$ for all $t \in \mathbb{Z}^+$ and $p_t \to 1$ as $t \to +\infty$, and (b) a discrete distribution $P_N$ over the agents $\mathcal{A}_1, \ldots, \mathcal{A}_N$ s.t. $P_N[n] \in [0, 1]$ for $n = 1, \ldots, N$ and $\sum_{n=1}^N P_N[n] = 1$. In iteration $t \geq 1$ of FTS, with probability $p_t$ (line 4 of Algorithm 1), $\mathcal{A}$ samples a function $f_t$ using its current GP posterior belief (1) and chooses $x_t = \arg\max_{x \in \mathcal{X}} f_t(x)$. With probability $1 - p_t$ (line 6), $\mathcal{A}$ firstly samples an agent $\mathcal{A}_n$ from $P_N$ and then chooses $x_t = \arg\max_{x \in \mathcal{X}} \hat{g}_n(x)$ where $\hat{g}_n(x) = \phi(x)^\top \omega_n$ corresponds to a sampled function from $\mathcal{A}_n$'s GP posterior belief with RFF approximation. Next, $x_t$ is queried to observe $y(x_t)$ and FTS proceeds to the next iteration $t + 1$.

---

**Algorithm 1** Federated Thompson Sampling (FTS)

---

1: **for** $t = 1, 2, \ldots, T$ **do**
2:     Sample $r$ from the uniform distribution over $[0, 1]$: $r \sim U(0, 1)$
3:     **if** $r \leq p_t$ **then**
4:        Sample $f_t \sim \mathcal{GP}(\mu_{t-1}(\cdot), \beta_t^2 \sigma_{t-1}^2(\cdot, \cdot))^1$ and choose $x_t = \arg\max_{x \in \mathcal{X}} f_t(x)$
5:     **else**
6:        Sample agent $\mathcal{A}_n$ from the distribution $P_N$ and choose $x_t = \arg\max_{x \in \mathcal{X}} \phi(x)^\top \omega_n$
7:     **end if**
8:     Query $x_t$ to observe $y(x_t)$ and update GP posterior belief (1) with $(x_t, y(x_t))$
9: **end for**

---

Interestingly, FTS (Algorithm 1) can be interpreted as a variant of *TS with a mixture of GPs*. That is, in each iteration $t$, we firstly sample a GP: The GP of $\mathcal{A}$ is sampled with probability $p_t$ while the GP of $\mathcal{A}_n$ is sampled with probability $(1 - p_t) P_N[n]$ for $n = 1, \ldots, N$. Next, we draw a function from the sampled GP whose maximizer is selected to be queried. As a result, $x_t$ follows the same distribution as the maximizer of the mixture of GPs and the mixture model gradually converges to the GP of $\mathcal{A}$ as $p_t \to 1$. The sequence $[p_t]_{t \in \mathbb{Z}^+}$ controls the degree of which information from the other agents is exploited, such that decreasing the value of this sequence encourages the utilization of such information. The distribution $P_N$ decides the preferences for different agents. A natural choice for $P_N$ is the uniform distribution $P_N[n] = 1/N$ for $n = 1, \ldots, N$ indicating equal preferences for all agents, which is a common choice when we have no knowledge regarding which agents are more similar to the target agent. In FTS, *stragglers*[2] can be naturally dealt with by simply assigning 0 to the corresponding agent $\mathcal{A}_n$ in the distribution $P_N$ such that $\mathcal{A}_n$ is never sampled (line 6 of Algorithm 1). Therefore, FTS is robust against communication failure which is a common issue in FL [35].

Since only one message $\omega_n$ is received from each agent *before* the beginning of FTS, once an agent $\mathcal{A}_n$ is sampled and its message $\omega_n$ is used (line 6 of Algorithm 1), we remove it from $P_N$ by setting the corresponding element to 0 and then re-normalize $P_N$. However, FTS can be easily generalized to allow $\mathcal{A}$ to receive information from each agent after every iteration (or every few iterations) such that every agent can be sampled multiple times. This more general setting requires more rounds of communication. In practice, FTS is expected to perform similarly in both settings when (a) the number $N$ of agents is large (i.e., a common assumption in FL), and (b) $P_N$ gives similar or equal preferences to all agents such that the probability of an agent being sampled more than once is small. Furthermore, this setting can be further generalized to encompass the scenario where multiple (even all) agents are concurrently performing optimization tasks using FTS. In this case, the information

received from $\mathcal{A}_n$ can be updated as $\mathcal{A}_n$ collects more observations, i.e., $t_n$ may increase as updated information is received from $\mathcal{A}_n$.

## 3.2 Comparison with Other BO Algorithms Modified for the FBO Setting

Although FTS is the first algorithm for the FBO setting, some algorithms for *meta-learning* in BO, such as *ranking-weighted GP ensemble* (RGPE) [16] and *transfer acquisition function* (TAF) [55], can be adapted to the FBO setting through a heuristic combination with RFF approximation. Meta-learning aims to use the information from previous tasks to accelerate the current task. Specifically, both RGPE and TAF use a separate GP surrogate to model the objective function of every agent (i.e., previous task) and use these GP surrogates to accelerate the current BO task. To modify both algorithms to suit the FBO setting, every agent $\mathcal{A}_n$ firstly applies RFF approximation to its own GP surrogate and passes the resulting parameters $\nu_n$ and $\Sigma_n^{-1}$ (Section 2) to the target agent $\mathcal{A}$. Next, after receiving $\nu_n$ and $\Sigma_n^{-1}$ from the other agents, $\mathcal{A}$ can use them to calculate the GP surrogate (with RFF approximation) of each agent (Section 2), which can then be plugged into the original RGPE/TAF algorithm.[3] However, unlike FTS, RGPE and TAF do not have theoretical convergence guarantee and thus lack an assurance to guarantee consistent performances in the presence of heterogeneous agents. Moreover, as we will analyze below and show in the experiments (Section 5), FTS outperforms both RGPE and TAF in a number of major aspects including communication efficiency, computational efficiency, and practical performance.

Firstly, regarding communication efficiency, both RGPE and TAF require $\nu_n$ and $\Sigma_n^{-1}$ (i.e., $M + M^2$ parameters) from each agent since both the posterior mean and variance of every agent are needed. Moreover, TAF additionally requires the incumbent (currently found maximum observation value) of every agent, which can further increase the risk of privacy leak. In a given experiment and for a fixed $M$, our FTS algorithm is superior in terms of communication efficiency since it only requires an $M$-dimensional vector $\omega_n$ from each agent, which is equivalent to standard FL using a linear model with $M$ parameters. Secondly, FTS is also advantageous in terms of computational efficiency: When $x_t$ is selected using $\omega_n$ from an agent, FTS only needs to solve the optimization problem of $x_t = \arg\max_{x \in \mathcal{X}} \phi(x)^\top \omega_n$ (line 6 of Algorithm 1), which incurs minimal computational cost;[4] when $x_t$ is selected by maximizing a sampled function from $\mathcal{A}$'s GP posterior belief (line 4 of Algorithm 1), this maximization step can also utilize the RFF approximation, which is computationally cheap. In contrast, for both RGPE and TAF, every evaluation of the acquisition function (i.e., to be maximized to select $x_t$) at an input $x \in \mathcal{X}$ requires calculating the posterior mean and variance using the GP surrogate of *every* agent at $x$. Therefore, their required computation in every iteration grows linearly in the number $N$ of agents and can thus become prohibitively costly when $N$ is large. We have also empirically verified this in our experiments (see Fig. 3d in Section 5.2).

## 4 Theoretical Results

In our theoretical analysis, since we allow the presence of heterogeneous agents (i.e., other agents with significantly different objective functions from the target agent), we do not aim to show that FTS achieves a faster convergence than standard TS and instead prove a convergence guarantee that is robust against heterogeneous agents. This is consistent with most works proving the convergence of FL algorithms [34, 35] and makes the theoretical results more applicable in general since the presence of heterogeneous agents is a major and inevitable challenge of FL and FBO. Note that we analyze FTS in the more general setting where communication is allowed before every iteration instead of only before the first iteration. However, as discussed in Section 3.1, FTS behaves similarly in both settings in the common scenario when $N$ is large and $P_N$ assigns similar probabilities to all agents. Theorem 1 below is our main theoretical result (see Appendix C for its proof):

**Theorem 1.** *Let $\gamma_t$ be the maximum information gain on $f$ from any set of $t$ observations. Let $\delta \in (0, 1)$, $\beta_t \triangleq B + \sigma\sqrt{2(\gamma_{t-1} + 1 + \log(4/\delta))}$, and $c_t \triangleq \beta_t(1 + \sqrt{2\log(|\mathcal{X}|t^2)})$ for all $t \in \mathbb{Z}^+$. Choose $[p_t]_{t \in \mathbb{Z}^+}$ as a monotonically increasing sequence satisfying $p_t \in (0, 1]$ for all $t \in \mathbb{Z}^+$, $p_t \to 1$ as $t \to +\infty$, and $(1 - p_t)c_t \leq (1 - p_1)c_1$ for all $t \in \mathbb{Z}^+ \setminus \{1\}$. With probability of at least $1 - \delta$,*

*the cumulative regret incurred by FTS is[5]*

$$R_T = \tilde{\mathcal{O}}\left((B + 1/p_1)\,\gamma_T\sqrt{T} + \sum\nolimits_{t=1}^{T}\psi_t\right)$$

*where $\psi_t \triangleq 2(1-p_t)\sum_{n=1}^{N}P_N[n]\Delta_{n,t}$ and $\Delta_{n,t} \triangleq \tilde{\mathcal{O}}(M^{-1/2}Bt_n^2 + B + \sqrt{\gamma_{t_n}} + \sqrt{M} + d_n + \sqrt{\gamma_t})$.*

Since $\gamma_T = \mathcal{O}((\log T)^{D+1})$ for the SE kernel [52], the first term in the upper bound is sublinear in $T$. Moreover, since the sequence of $[p_t]_{t\in\mathbb{Z}^+}$ is chosen to be monotonically increasing and goes to 1 when $t \to \infty$, $1 - p_t$ goes to 0 asymptotically. Therefore, the second term in the upper bound also grows sublinearly.[6] For example, if $[p_t]_{t\in\mathbb{Z}^+}$ is chosen such that $1 - p_t = \mathcal{O}(1/\sqrt{t})$, $\sum_{t=1}^{T}\psi_t = \tilde{\mathcal{O}}(\sqrt{T})$. As a result, FTS achieves *no regret* asymptotically regardless of the difference between the target agent and the other agents, which is a highly desirable property for FBO in which the heterogeneity among agents is a prominent challenge. Such a robust regret upper bound is achieved because we upper-bound the worst-case error for *any* set of agents (i.e., any set of values of $d_n$ and $t_n$ for $n = 1, \ldots, N$) in our proof. The robust nature of the regret upper bound can be reflected in its dependence on the sequence $[p_t]_{t\in\mathbb{Z}^+}$ as well as on $d_n$ and $t_n$. When the value of the sequence $[p_t]_{t\in\mathbb{Z}^+}$ is small, i.e., when the information from the other agents is exploited more (Section 3.1), the worst-case error due to more utilization of these information is also increased. This is corroborated by Theorem 1 since smaller values of $p_t$ increase the regret upper bound through the terms $1/p_1$ and $(1 - p_t)$ in $\psi_t$. Theorem 1 also shows that the regret bound becomes worse with larger values of $d_n$ and $t_n$ because a larger $d_n$ increases the difference between the objective functions of $\mathcal{A}_n$ and $\mathcal{A}$, and more observations from $\mathcal{A}_n$ (i.e., larger $t_n$) also loosens the upper bound since for a fixed $d_n$, a larger number of observations increases the worst-case error by accumulating the individual errors.[7]

The dependence of the regret upper bound (through $\Delta_{n,t}$) on the number $M$ of random features is particularly interesting due to the interaction between two opposing factors. Firstly, the $M^{-1/2}Bt_n^2$ term arises since better approximation quality of the agent's GP surrogates (i.e., larger $M$) improves the performance. However, the $\sqrt{M}$ term suggests the presence of another factor with an opposite effect. This results from the need to upper-bound the distance between the $M$-dimensional Gaussian random variable $\omega_n$ and its mean $\nu_n$ (2), which grows at a rate of $\mathcal{O}(\sqrt{M})$ (Lemma 3 in Appendix C). Taking the derivative of both terms w.r.t. $M$ reveals that the regret bound is guaranteed to become tighter with an increasing $M$ (i.e., the effect of the $M^{-1/2}Bt_n^2$ term dominates more) when $t_n$ is sufficiently large, i.e., when $t_n = \Omega(\sqrt{M/B})$. An intuitive explanation of this finding, which is verified in our experiments (Section 5.1), is that the positive effect (i.e., a tighter regret bound) of better RFF approximation from a larger $M$ is amplified when more observations are available (i.e., $t_n$ is large). In contrast, when $t_n$ is small, minimal information is offered by agent $\mathcal{A}_n$ and increasing the quality of RFF approximation thus only leads to marginal or negligible improvement in the performance. The practical implication of this insight is that when the other agents only have a small number of observations, it is not recommended to use a large number of random features since it requires a larger communication bandwidth (Section 3.2) yet is unlikely to improve the performance.

## 5  Experiments and Discussion

We firstly use synthetic functions to investigate the behavior of FTS. Next, using 3 real-world experiments, we demonstrate the effectiveness of FTS in terms of communication efficiency, computational efficiency, and practical performance. Since it has been repeatedly observed that the theoretical choice of $\beta_t$ that is used to establish the confidence interval is overly conservative [2, 52], we set it to a constant: $\beta_t = 1.0$. As a result, $c_t$ (Theorem 1) grows slowly (i.e., logarithmically) and we thus do not explicitly check the validity of the condition $(1 - p_t)c_t \leq (1 - p_1)c_1$ for all $t \in \mathbb{Z}^+ \setminus \{1\}$. All error bars represent standard errors. For simplicity, we focus on the simple setting here where communication happens only before the beginning of FTS (Section 3.1). In Appendix D.2.1, we also evaluate the performance of FTS in the most general setting where the other agents are also performing optimization tasks such that they may collect more observations between different rounds

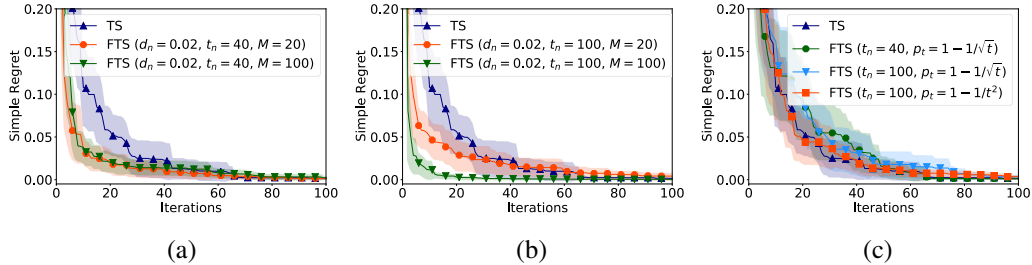

Figure 1: Simple regret in synthetic experiments showing the effect of $M$ when $t_n$ is (a) small and (b) large, and (c) the performance when $d_n = 1.2$ is large. Each curve is averaged over $5$ randomly sampled functions from a GP and $5$ random initializations of $1$ input for each function.

of communication (i.e., increasing $t_n$). The results (Fig. 4 in Appendix D.2.1) show consistent performances of FTS in both settings. More experimental details and results are deferred to Appendix D due to space constraint.

## 5.1 Optimization of Synthetic Functions

In synthetic experiments, the objective functions are sampled from a GP (i.e., defined on a 1-D discrete domain within $[0, 1]$) using the SE kernel and scaled into the range $[0, 1]$. We fix the total number of agents as $N = 50$ and vary $d_n$, $t_n$, and $M$ to investigate their effects on the performance. We use the same $d_n$ and $t_n$ for all agents for simplicity. We choose $P_N$ to be uniform: $P_N[n] = 1/N$ for $n = 1, \ldots, N$ and choose the sequence $[p_t]_{t \in \mathbb{Z}^+}$ as $p_t = 1 - 1/\sqrt{t}$ for all $t \in \mathbb{Z}^+ \setminus \{1\}$ and $p_1 = p_2$. Figs. 1a and b show that when $d_n = 0.02$ is small, FTS is able to perform better than TS. Intuitively, the performance advantage of FTS results from its ability to exploit the additional information from the other agents to reduce the need for exploration. These results also reveal that when $t_n$ of every agent is small (Fig. 1a), the effect of $M$ is negligible; on the other hand, when $t_n$ is large (Fig. 1b), increasing $M$ leads to evident improvement in the performance. This corroborates our theoretical analysis (Section 4) stating that when $t_n$ is large, increasing the value of $M$ is more likely to tighten the regret bound and hence improve the performance. Moreover, comparing the green curves in Figs. 1a and b shows that when the other agents' objective functions are similar to the target agent's objective function (i.e., $d_n = 0.02$ is small) and the RFF approximation is accurate (i.e., $M = 100$ is large), increasing the number of observations from the other agents ($t_n = 100$ vs. $t_n = 40$) improves the performance. Lastly, Fig. 1c verifies FTS's theoretically guaranteed robustness against heterogeneous agents (Section 4) since it shows that even when all other agents are heterogeneous (i.e., every $d_n = 1.2$ is large), the performances of FTS are still comparable to that of standard TS. Note that Fig. 1c demonstrates a potential limitation of our method, i.e., in this scenario of heterogeneous agents, FTS may converge slightly slower than TS if $p_t$ does not grow sufficiently fast. However, the figure also shows that making $p_t$ grow faster (i.e., making the effect of the other agents decay faster) allows FTS to match the performance of TS in this adverse scenario (red curve).

## 5.2 Real-world Experiments

For real-world experiments, we use 3 datasets generated in federated settings that naturally contain heterogeneous agents [51]. Firstly, we use a landmine detection dataset in which the landmine fields are located at two different terrains [58]. Next, we use two activity recognition datasets collected using Google glasses [44] and mobile phone sensors [1], both of which contain heterogeneous agents since cross-subject heterogeneity has been a major challenge for human activity recognition [44]. We compare our FTS algorithm with standard TS (i.e., no communication with other agents), RGPE, and TAF. Note that RGPE and TAF are meta-learning algorithms for BO and are hence not specifically designed for the FBO setting (Section 3.2).

**Landmine Detection.** This dataset includes 29 landmine fields. For each field, every entry in the dataset consists of 9 features and a binary label indicating whether the corresponding location contains landmines. The task of every field is to tune 2 hyperparameters of an SVM classifier (i.e., RBF kernel parameter in $[0.01, 10]$ and L2 regularization parameter in $[10^{-4}, 10]$) that is used to predict whether

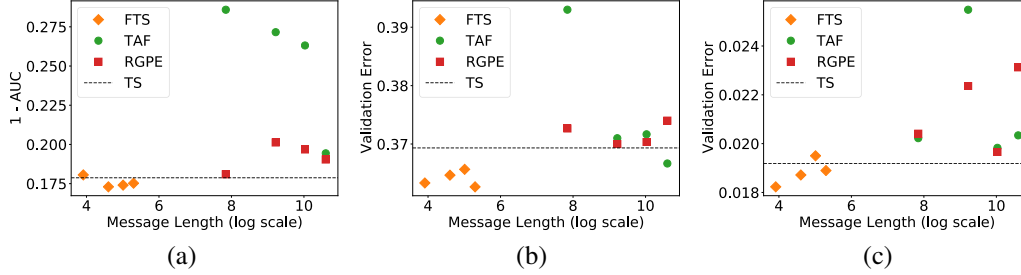

(a)                    (b)                    (c)

Figure 2: Best performance after 50 iterations (vertical) vs. the length of the message (i.e., the number of parameters) communicated from each agent (horizontal) for the (a) landmine detection, (b) Google glasses, and (c) mobile phone sensors experiments. The more to the *bottom left*, the better the performance and the less the required communication. The results for every method correspond to $M = 50, 100, 150, 200$, respectively. Every result is averaged over 6 different target agents and each target agent is averaged over 5 different initializations of 3 randomly selected inputs.

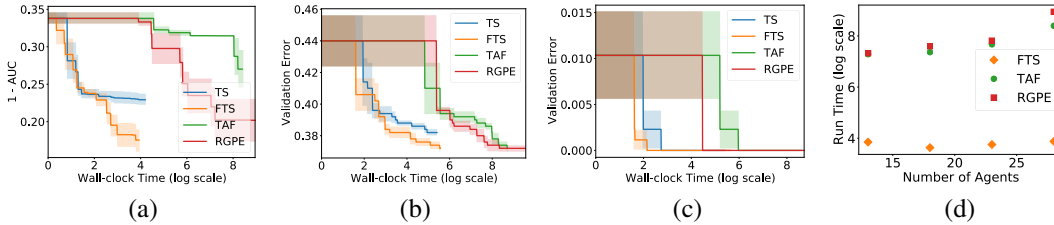

(a)                (b)                (c)                (d)

Figure 3: Best performance observed vs. run time (seconds) for the (a) landmine detection, (b) Google glasses, and (c) mobile phone sensors experiments, in which *FTS converges faster than other methods*. These results correspond to the first (of the 6) target agent used in each experiment in Fig. 2 with $M = 100$ and are averaged over 5 random initializations of 3 inputs.[8] Every method is run for 50 iterations. (d) Total runtime vs. the number of agents for the landmine detection experiment.

a location contains landmines. We fix one of the landmine fields as the target agent and the remaining $N = 28$ fields as the other agents, each of whom has completed a BO task of $t_n = 50$ iterations.

**Activity Recognition Using Google Glasses.** This dataset contains sensor measurements from Google glasses worn by 38 participants. Every agent attempts to use 57 features, which we have extracted from the corresponding participant's measurements, to predict whether the participant is eating or performing other activities. Every agent uses *logistic regression* (LR) for activity prediction and needs to tune 3 hyperparameters of LR: batch size ($[20, 60]$), L2 regularization parameter ($[10^{-6}, 1]$), and learning rate ($[0.01, 0.1]$). We fix one of the participants as the target agent and all other $N = 37$ participants as the other agents, each of whom possesses $t_n = 50$ BO observations.

**Activity Recognition Using Mobile Phone Sensors.** This dataset consists of mobile phone sensor measurements from 30 subjects performing 6 activities. Each agent attempts to tune the hyperparameters of a subject's activity prediction model whose input includes 561 features and output is one of the 6 activity classes. The activity prediction model and tuned hyperparameters, as well as their ranges, are the same as that in the Google glasses experiment. We again fix one of the subjects as the target agent and all other $N = 29$ subjects as the other agents with $t_n = 50$ observations each.

For all experiments, we set $P_N$ to be uniform: $P_N[n] = 1/N, \forall n = 1, \ldots, N$, and $p_t = 1 - 1/t^2$ for all $t \in \mathbb{Z}^+ \setminus \{1\}$ and $p_1 = p_2$. We use validation error as the performance metric for the two activity recognition experiments, and use *area under the receiver operating characteristic curve* (AUC) to measure the performance of the landmine detection experiment since this dataset is extremely imbalanced (i.e., only $6.2\%$ of all locations contain landmines). We repeat every experiment 6 times with each time treating one of the first 6 agents as the target agent. Fig. 2 shows the (averaged) best performance after 50 iterations of different methods (vertical axis) as well as their required number of parameters to be passed from each agent (horizontal axis). FTS outperforms both RGPE and TAF in terms of both the *performance metric* and the *communication efficiency*. Note that this comparison is unfair for FTS since FTS is much more computationally efficient than RGPE and TAF (Section 3.2) such that it completes 50 iterations in significantly shorter time (Fig. 3). Fig. 3 plots the best performance achieved vs. the run time of different algorithms with the first agent treated as

the target agent; refer to Appendix D.2.3 for the results of the other agents.[8] Fig. 3 shows that FTS achieves the fastest convergence among all methods and showcases the advantage of FTS over RGPE and TAF in terms of *computational efficiency* (Section 3.2). Overall, the consistent performance advantage of FTS across all real-world experiments is an indication of its practical robustness, which may be largely attributed to its robust theoretical convergence guarantee ensuring its consistent performance even in the presence of heterogeneous agents (Section 4). Furthermore, we also use the landmine detection experiment to illustrate the scalability of our method w.r.t. the number $N$ of agents. The results (Fig. 3d) show that increasing $N$ has minimal effect on the runtime of FTS yet leads to growing computational cost for RGPE and TAF. This verifies the relevant discussion at the end of Section 3.2.

## 6   Related Works

Since its recent introduction in [39], FL has gained tremendous attention mainly due to its prominent practical relevance in the collaborative training of ML models such as DNNs [39] or decision tree-based models [31, 32]. Meanwhile, efforts have also been made to derive theoretical convergence guarantees for FL algorithms [34, 35]. Refer to recent surveys [24, 30, 33] for more comprehensive reviews of FL. TS [54] has been known as a highly effective practical technique for multi-armed bandit problems [4, 47]. The Bayesian regret [46] and frequentist regret [9] of TS in BO have both been analyzed and TS has been shown to perform effectively in BO problems such as high-dimensional BO [40]. The theoretical analysis in this work has adopted techniques used in the works of [9, 40]. Our algorithm is also related to multi-fidelity BO [12, 25, 42, 56, 62, 63] which has the option to query low-fidelity functions. This is analogous to our algorithm allowing the target agent to use the information from the other agents for query selection and the similarity between an agent and the target agent can be interpreted as a measure of fidelity. Moreover, our algorithm also bears similarity to parallel/distributed BO algorithms [10, 13, 14], especially those based on TS [17, 26]. However, there are fundamental differences: For example, they usually optimize a single objective function whereas we need to consider possibly heterogeneous objective functions from different agents. On the other hand, BO involving multiple agents with possibly different objective functions has been studied from the perspective of game theory by the works of [11, 48]. As discussed in Section 3.2, some works on meta-learning for BO [16, 55], which study how information from other related BO tasks is used to accelerate the current BO task, can be adapted to the FBO setting. However, these works do not provide theoretical convergence guarantee nor tackle the issues of avoiding the transmission of raw data and achieving efficient communication. Moreover, their adapted variants for FBO have been shown to be outperformed by our FTS algorithm in various major aspects including communication efficiency, computational efficiency, and practical performance (Section 5.2).

## 7   Conclusion and Future Works

This paper introduces the first algorithm for the FBO setting called FTS which addresses some key challenges in FBO in a principled manner. We theoretically show its convergence guarantee which is robust against heterogeneous agents, and empirically demonstrate its communication efficiency, computational efficiency, and practical effectiveness using three real-world experiments. As a future work, we plan to explore techniques to automatically optimize the distribution $P_N$ used by FTS to sample agents by learning the similarity between each agent and the target agent (i.e., the fidelity of each agent). Other than the RFF approximation used in this work, other approximation techniques for GP (such as those based on inducing points [5, 6, 7, 8, 18, 19, 20, 21, 23, 37, 38, 41, 53, 57, 59, 60]) may also be used to derive the parameters to be exchanged between agents, which is worth exploring in future works. Moreover, in our experiments, the hyperparameters of the target agent's GP is learned by maximizing the marginal likelihood; it would be interesting to explore whether the GP hyperparameters can also be shared among the agents, which can potentially facilitate better collaboration. Furthermore, our current algorithm is only able to avoid sharing of raw data and may hence be susceptible to advanced privacy attacks. So, it would be interesting to incorporate the state-of-the-art privacy-preserving techniques into our algorithm such as differential privacy [15] which has been applied to BO by the works of [27, 29]. We will consider incentivizing collaboration in FBO [50] and generalizing FBO to nonmyopic BO [28, 36] and high-dimensional BO [22] settings.

## Broader Impact

Since the setting of our FBO is similar to that of FL, our work inherits a number of potential broader impacts of FL. We will analyze here the potential impacts of our work in the scenario where the individual agents are edge devices such as mobile phones since it is a major area of application for FBO and FL.

Specifically, since our algorithm can be used to improve the efficiency of black-box optimization tasks performed by mobile phones, it has the potential of dramatically improving the efficacy and function of various applications for the user (e.g., the smart keyboard example that we mentioned in Section 1), which will enhance their user experience and productivity. On the other hand, some negative impacts of FL also need to be considered when promoting the widespread application of our work. For example, although our work is able to prevent exchanging the raw data in the same way as standard FL, advanced privacy attack methods (e.g., inference attack) may still incur risks of privacy leak for FL and thus FBO. Preventing this risk through principled privacy protection techniques (e.g., differential privacy) is important for the widespread adoption of FL and FBO algorithms and hence represents an interesting and promising direction for future research.

## Acknowledgments and Disclosure of Funding

This research/project is supported by A*STAR under its RIE2020 Advanced Manufacturing and Engineering (AME) Industry Alignment Fund – Pre Positioning (IAF-PP) (Award A19E4a0101).

## Footnotes

[1] We will define $\beta_t$ in Theorem 1 (Section 4).

[2] Stragglers refer to those agents whose information is not received by the target agent [35].

[3]Refer to [16] and [55] for more details about RGPE and TAF, respectively.

[4]We use the DIRECT method for this optimization problem, which, for example, takes on average 0.76 seconds per iteration in the landmine detection experiment (Section 5.2).

[5]The $\tilde{\mathcal{O}}$ notation ignores all logarithmic factors.

[6]Note that for the SE kernel, $\sqrt{\gamma_t}$ in $\Delta_{n,t}$ is logarithmic in $t$: $\sqrt{\gamma_t} = \mathcal{O}((\log t)^{(D+1)/2})$.

[7]In the most general setting where $\mathcal{A}_n$ may collect more observations between different rounds of communication such that $t_n$ may increase (Section 3.1), $1 - p_t$ can decay faster to preserve the no-regret convergence.

[8]We cannot average the results across different agents since their output scales vary significantly.

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
