[Supplementary Material]

## A Construction of Random Fourier Features

As mentioned in Section 2, in this work, we focus on the widely used *squared exponential* (SE) kernel: $k(x, x') = \sigma_0^2 \exp(-\|x - x'\|_2^2 / (2l^2))$ in which $l$ is the length scale and $\sigma_0^2$ is the signal variance. $\sigma_0^2 = 1$ is usually the default value, which we use in all experiments. We construct the random features following the work of [43]. Specifically, for the SE kernel with length scale $l$, the spectral density follows a $D$-dimensional Gaussian distribution: $p(s) = \mathcal{N}(0, \frac{1}{l^2} I_{D \times D})$. To begin with, we draw $M$ independent samples of $\{s_i\}_{i=1,...,M}$ from $p(s)$ (every $s_i$ is a $D$-dimensional vector), and $M$ independent samples of $\{b_i\}_{i=1,...,M}$ from the uniform distribution over $[0, 2\pi]$ (every $b_i$ is a scalar). Next, for an input $x$, the corresponding $M$-dimensional random features (basis functions) can be constructed as $\phi(x)^\top = [\sqrt{2/M} \cos(s_i^\top x + b_i)]_{i=1,...,M}$. Each set of random features $\phi(x)$ is then normalized such that $\|\phi(x)\|_2^2 = \sigma_0^2$. As a result, sharing the random features $\phi(x), \forall x \in \mathcal{X}$ among all agents (Section 2) can be achieved by simply sharing the parameters $\{s_i\}_{i=1,...,M}$ and $\{b_i\}_{i=1,...,M}$. This is easily achievable since it is equivalent to sharing the parameters of the first layer of a neural network model with $M$ units in the hidden layer, in which $\{s_i\}_{i=1,...,M}$ are the weights (which form a $D \times M$-dimensional weight matrix) and $\{b_i\}_{i=1,...,M}$ are the biases.

## B GP Posterior/Predictive Belief with Random Fourier Features Approximation

Here we derive the expressions of the posterior/predictive mean and variance of a GP with random Fourier features (RFF) approximation (Section 2). Recall that we have defined $\Phi(X_t) = [\phi(x_1), \ldots, \phi(x_t)]^\top$ which is a $t \times M$-dimensional matrix.

With the RFF approximation, the kernel function is approximated by $k(x, x') \approx \phi(x)^\top \phi(x')$. Define $\hat{K}_t = [\phi(x_{t'})^\top \phi(x_{t''})]_{t',t''=1,...,t} = \Phi(X_t) \Phi(X_t)^\top$ and $\hat{k}_t(x) = [\phi(x)^\top \phi(x_{t'})]_{t'=1,...,t}^\top = \Phi(X_t) \phi(x)$, which are analogous to $K_t$ and $k_t(x)$ in (1) with the kernel values $k(x, x')$ replaced by the approximate kernel values $\phi(x)^\top \phi(x')$. With these definitions, we have that

$$
\begin{aligned}
\Phi(X_t)^\top \left[ \hat{K}_t + \sigma^2 I \right] &= \Phi(X_t)^\top \left[ \Phi(X_t) \Phi(X_t)^\top + \sigma^2 I \right] \\
&= \Phi(X_t)^\top \Phi(X_t) \Phi(X_t)^\top + \sigma^2 \Phi(X_t)^\top \\
&= \left[ \Phi(X_t)^\top \Phi(X_t) + \sigma^2 I \right] \Phi(X_t)^\top \\
&= \Sigma_t \Phi(X_t)^\top.
\end{aligned}
\tag{4}
$$

Multiplying both sides by $\Sigma_t^{-1}$ from the left and $(\hat{K}_t + \sigma^2 I)^{-1}$ from the right, we get

$$
\Sigma_t^{-1} \Phi(X_t)^\top = \Phi(X_t)^\top (\hat{K}_t + \sigma^2 I)^{-1}.
\tag{5}
$$

Then multiplying both sides by $\phi(x)^\top$ from the left and $y_t$ from the right, we get

$$
\begin{aligned}
\hat{\mu}_t(x) = \phi(x)^\top \nu_t = \phi(x)^\top \Sigma_t^{-1} \Phi(X_t)^\top y_t &= \phi(x)^\top \Phi(X_t)^\top (\hat{K}_t + \sigma^2 I)^{-1} y_t \\
&= \hat{k}_t(x)^\top (\hat{K}_t + \sigma^2 I)^{-1} y_t,
\end{aligned}
\tag{6}
$$

which proves that the expression of the approximate posterior mean with RFF approximation: $\hat{\mu}_t(x) = \phi(x)^\top \nu_t$ matches the expression of the posterior mean of standard GP without RFF approximation, except that the kernel values $k(x, x')$ are replaced by the approximate kernel values $\phi(x)^\top \phi(x')$.

Next, we derive the expression of the approximate posterior variance. Making use of the matrix inversion lemma, we get

$$
\begin{aligned}
\hat{\sigma}_t^2(x) = \sigma^2 \phi(x)^\top \Sigma_t^{-1} \phi(x) &= \sigma^2 \phi(x)^\top (\Phi(X_t)^\top \Phi(X_t) + \sigma^2 I)^{-1} \phi(x) \\
&= \sigma^2 \phi(x)^\top \left[ \frac{1}{\sigma^2} I - \frac{1}{\sigma^2} \Phi(X_t)^\top \left( I + \Phi(X_t) \frac{1}{\sigma^2} \Phi(X_t)^\top \right)^{-1} \Phi(X_t) \frac{1}{\sigma^2} \right] \phi(x) \\
&= \phi(x)^\top \phi(x) - \phi(x)^\top \Phi(X_t)^\top \left( \sigma^2 I + \Phi(X_t)^\top \Phi(X_t) \right)^{-1} \Phi(X_t) \phi(x) \\
&= \hat{k}(x, x) - \hat{k}_t(x)^\top \left( \hat{K}_t + \sigma^2 I \right)^{-1} \hat{k}_t(x),
\end{aligned}
\tag{7}
$$

which gives the expression of the approximate posterior variance: $\hat{\sigma}_t^2(x) = \sigma^2 \phi(x)^\top \Sigma_t^{-1} \phi(x)$. To conclude, Equations (6) and (7) prove that the expressions of the GP posterior mean and variance with RFF approximation given in Section 2 (in the paragraph after Equation (3)) match the corresponding expressions of standard GP posterior mean and variance without RFF approximation (1), except that the original kernel values (i.e., $k(x, x')$) are replaced by the corresponding approximate kernel values (i.e., $\phi(x)^\top \phi(x')$).

## C   Proof of Theorem 1

As mentioned in Section 4, we analyze our FTS algorithm in the more general setting in which a message can be received from each agent $\mathcal{A}_n$ before every iteration $t$, instead of only before the first iteration. Therefore, throughout our theoretical analysis, we use $\omega_{n,t}$, instead of $\omega_n$, to denote the message received from agent $\mathcal{A}_n$ before iteration $t$. Similarly, we use $\hat{g}_{n,t}$, instead of $\hat{g}_n$, to denote the corresponding sampled function from agent $\mathcal{A}_n$ with RFF approximation in iteration $t$, obtained using $\omega_{n,t}$: $\hat{g}_{n,t}(x) = \phi(x)^\top \omega_{n,t}, \forall x \in \mathcal{X}$. Note that our theoretical analysis and results also hold in the most general setting where every agent $\mathcal{A}_n$ may collect more observations between different rounds of communication, in which the only difference is that every $t_n, \forall n = 1, \ldots, N$ may increase over different iterations.

Define $\mathcal{F}_t$ as the filtration containing agent $\mathcal{A}$'s history of selected inputs and observed outputs up to iteration $t$. Let $\delta \in (0, 1)$, we have defined in Theorem 1 that $\beta_t = B + \sigma \sqrt{2(\gamma_{t-1} + 1 + \log(4/\delta)}$ and $c_t = \beta_t(1 + \sqrt{2\log(|\mathcal{X}|t^2)})$ for all $t \in \mathbb{Z}^+$. Clearly, both $\beta_t$ and $c_t$ are increasing in $t$. Denote by $A_t$ the event that agent $\mathcal{A}$ chooses $x_t$ by maximizing a sampled function from its own GP posterior belief (i.e., $x_t = \arg\max_{x \in \mathcal{X}} f_t(x)$, as in line 4 of Algorithm 1), which happens with probability $p_t$; denote by $B_t$ the event that $\mathcal{A}$ chooses $x_t$ by maximizing the sampled function from any other agent $\mathcal{A}_1, \ldots, \mathcal{A}_N$ (line 6 of Algorithm 1), which happens with probability $(1 - p_t)$; denote by $B_{t,n}$ the event that $\mathcal{A}$ chooses $x_t$ by maximizing the sampled function of agent $\mathcal{A}_n$ using RFF approximation (i.e., $x_t = \arg\max_{x \in \mathcal{X}} \hat{g}_{n,t}(x)$), which happens with probability $(1 - p_t) \times P_N[n]$.

To begin with, we define two high-probability events through the following lemmas.

**Lemma 1.** *Let $\delta \in (0, 1)$. Define $E^f(t)$ as the event that $|\mu_{t-1}(x) - f(x)| \leq \beta_t \sigma_{t-1}(x)$ for all $x \in \mathcal{X}$. We have that $\mathbb{P}\left[E^f(t)\right] \geq 1 - \delta/4$ for all $t \geq 1$.*

Lemma 1 quantifies the concentration of the function $f$ around its posterior mean and its proof follows directly from Theorem 2 of the work of [9] by using an error probability of $\delta/4$.

**Lemma 2.** *Define $E^{f_t}(t)$ as the event that $|f_t(x) - \mu_{t-1}(x)| \leq \beta_t \sqrt{2\log(|\mathcal{X}|t^2)}\sigma_{t-1}(x)$. We have that $\mathbb{P}\left[E^{f_t}(t)|\mathcal{F}_{t-1}\right] \geq 1 - 1/t^2$ for any possible filtration $\mathcal{F}_{t-1}$.*

Lemma 2 illustrates how concentrated a sampled function $f_t$ from $f$ is around its posterior mean and is a simpler version of Lemma 5 of the work of [9]. Specifically, we have assumed a discrete domain, whereas the work of [9] deals with a compact domain. Note that both events $E^f(t)$ and $E^{f_t}(t)$ are $\mathcal{F}_{t-1}$-measurable.

Next, we define a set of inputs at every iteration $t$ called *saturated points*, which represents the set of "bad" inputs at iteration $t$. These inputs are "bad" in the sense that the function values at these inputs have relatively large difference from the value of the global maximum of $f$. In the subsequent proof, we will lower-bound the probability that the selected input $x_t$ is *unsaturated*, which will be a critical step in the proof.

**Definition 1.** *Define the set of saturated points at iteration $t$ as*

$$S_t = \{x \in \mathcal{X} : \Delta(x) > c_t \sigma_{t-1}(x)\}$$

*in which $\Delta(x) = f(x^*) - f(x)$ and $x^* = \arg\max_{x \in \mathcal{X}} f(x)$.*

Note that from this definition, $x^*$ is always unsaturated since $\Delta(x) = f(x^*) - f(x^*) = 0 < c_t \sigma_{t-1}(x^*)$ for all $t \geq 1$. Also note that $S_t$ is $\mathcal{F}_{t-1}$-measurable.

The next lemma bounds the deviation of the sampled function $\hat{g}_{n,t}(x)$ from agent $\mathcal{A}_n$'s GP posterior belief with RFF approximation around its posterior mean $\hat{\mu}_{n,t}(x)$, whose proof is based on that of Lemma 11 of [40].

**Lemma 3.** *Given $\delta \in (0, 1)$. We have that for all agents $\mathcal{A}_n, \forall n = 1, \ldots, N$, all $x \in \mathcal{X}$ and all $t \geq 1$, with probability of at least $1 - \delta/4$*

$$|\hat{\mu}_{n,t}(x) - \hat{g}_{n,t}(x)| \leq \sqrt{2 \log \frac{2\pi^2 t^2 N}{3\delta} + M}.$$

*Proof.* Recall from Section 2 that the sampled function $\hat{g}_{n,t}$ is obtained by firstly sampling $\omega_{n,t} \sim \mathcal{N}(\nu_{n,t}, \sigma^2 \Sigma_{n,t}^{-1})$, and then setting $\hat{g}_{n,t}(x) = \phi(x)^\top \omega_{n,t}, \forall x \in \mathcal{X}$. Moreover, we have shown in Section 2 that $\hat{\mu}_{n,t}(x) = \phi(x)^\top \nu_{n,t}$. Denote $\omega_{n,t} = \nu_{n,t} + \sigma \Sigma_{n,t}^{-1/2} z$, in which $z \sim \mathcal{N}(0, I)$ is the $M \times 1$-dimensional standard Gaussian distribution. We have that

$$\begin{aligned}
|\hat{\mu}_{n,t}(x) - \hat{g}_{n,t}(x)|^2 &= |\phi(x)^\top \nu_{n,t} - \phi(x)^\top (\nu_{n,t} + \sigma \Sigma_{n,t}^{-1/2} z)|^2 \\
&= |\sigma \phi(x)^\top \Sigma_{n,t}^{-1/2} z|^2 \\
&\leq \sigma^2 \left\| \phi(x)^\top \Sigma_{n,t}^{-1/2} \right\|_2^2 \|z\|_2^2 \\
&= \sigma^2 \phi(x)^\top \Sigma_{n,t}^{-1} \phi(x) \|z\|_2^2 \\
&= \hat{\sigma}_{n,t}^2(x) \|z\|_2^2 \leq \|z\|_2^2,
\end{aligned} \qquad (8)$$

in which we have made use of the assumption w.l.o.g. that the posterior variance is upper-bounded by $1$ in the last inequality. Next, making use of the concentration of chi-squared distribution: $\mathbb{P}(\|z\|_2^2 \geq M + 2\lambda) \leq \exp(-\lambda)$ [40], we have that with probability of at least $1 - \frac{3\delta}{2\pi^2 t^2 N}$,

$$\|z\|_2^2 \leq M + 2 \log \frac{2\pi^2 t^2 N}{3\delta}. \qquad (9)$$

Taking a union bound over all agents $\mathcal{A}_1, \ldots, \mathcal{A}_N$ and over all $t \geq 1$ completes the proof. $\qquad \square$

The following lemma uniformly upper-bounds the difference between agent $\mathcal{A}_n$'s objective function $g_n$ and sampled function $\hat{g}_{n,t}$ from its GP posterior belief with RFF approximation.

**Lemma 4.** *Given any $\delta \in (0, 1)$. For agent $\mathcal{A}_n$'s sampled function $\hat{g}_{n,t}$ from its GP posterior belief with RFF approximation, we have that for all agents $\mathcal{A}_n, \forall n = 1, \ldots, N$, all $x \in \mathcal{X}$ and all $t \geq 1$, with probability of at least $1 - \delta/2$,*

$$|\hat{g}_{n,t}(x) - g_n(x)| \leq \tilde{\Delta}_{n,t},$$

*where $\beta_t' = B + \sigma \sqrt{2(\gamma_{t-1} + 1 + \log(8N/\delta))}$, and*

$$\tilde{\Delta}_{n,t} \triangleq \varepsilon \frac{(t_n + 1)^2}{\sigma^2} \left( B + \sqrt{2 \log \left( \frac{4\pi^2 t^2 N}{3\delta} \right)} \right) + \beta_{t_n+1}' + \sqrt{2 \log \frac{2\pi^2 t^2 N}{3\delta} + M}.$$

*Proof.* Recall that $\varepsilon$ is the accuracy of the RFF approximation, $t_n$ is the number of iterations that agent $\mathcal{A}_n$ has completed in its own BO task when it passes information to $\mathcal{A}$, $M$ is the number of random features used in the RFF approximation. Denote by $\hat{\mu}_{n,t}(x)$ and $\mu_{n,t}(x)$ ($\hat{\sigma}_{n,t}(x)$ and $\sigma_{n,t}(x)$) the posterior mean (standard deviation) at $x$ of agent $\mathcal{A}_n$'s GP after running its own BO task for $t_n$ iterations with and without the RFF approximation respectively.

We have that for all $x \in \mathcal{X}$, all agents $\mathcal{A}_n$ and all $t \geq 1$, with probability of at least $1 - \delta/8$,

$$|\mu_{n,t}(x) - \hat{\mu}_{n,t}(x)| \leq \varepsilon \frac{(t_n + 1)^2}{\sigma^2} \left( B + \sqrt{2 \log \left( \frac{4\pi^2 t^2 N}{3\delta} \right)} \right), \qquad (10)$$

which can be proved by following the proof of Theorem 5 in the work of [40] by substituting the error probability of $\frac{3\delta}{4\pi^2 t^2 N}$, and taking a union bound over all agents and all $t \geq 1$. Next, making use of Lemma 1 (replacing $f$ by $g_n$, and $\delta/4$ by $\delta/(8N)$), we get

$$|\mu_{n,t}(x) - g_n(x)| \leq \beta_{t_n+1}' \sigma_{n,t}(x) \leq \beta_{t_n+1}', \qquad (11)$$

which holds for all $x \in \mathcal{X}$, agents $\mathcal{A}_n$ and $t_n \geq 1$, with probability of at least $1 - \delta/8$. The last inequality follows from our assumption w.l.o.g. that the posterior variance is upper-bounded by $1$.

Combining the two equations above and making use of Lemma 3 completes the proof. $\qquad \square$

The next lemma shows a uniform upper bound on the difference between the sampled function $f_t$ of agent $\mathcal{A}$ and that of agent $\mathcal{A}_n$ with RFF approximation $(\hat{g}_{n,t})$.

**Lemma 5.** *At iteration $t$, conditioned on the events $E^f(t)$ and $E^{f_t}(t)$, we have that for all agents $\mathcal{A}_n, \forall n = 1, \ldots, N$ and for all $x \in \mathcal{X}$ with probability $\geq 1 - \delta/2$*

$$|\hat{g}_{n,t}(x) - f_t(x)| \leq \Delta_{n,t},$$

*in which*

$$\Delta_{n,t} \triangleq \varepsilon \frac{(t_n+1)^2}{\sigma^2} \left( B + \sqrt{2\log\left(\frac{4\pi^2 t^2 N}{3\delta}\right)} \right) + \beta'_{t_n+1} + \sqrt{2\log\frac{2\pi^2 t^2 N}{3\delta}} + M + d_n + c_t.$$

*Proof.* Firstly, note that since we condition on both events $E^f(t)$ and $E^{f_t}(t)$, we have that for all $x \in \mathcal{X}$ and all $t \geq 1$

$$
\begin{aligned}
|f(x) - f_t(x)| &\leq |f(x) - \mu_{t-1}(x)| + |\mu_{t-1}(x) - f_t(x)| \\
&= \beta_t \sigma_{t-1}(x) + \beta_t \sqrt{2\log(|\mathcal{X}|t^2)}\sigma_{t-1}(x) = c_t \sigma_{t-1}(x)
\end{aligned}
\tag{12}
$$

Next,

$$
\begin{aligned}
|\hat{g}_{n,t}(x) - f_t(x)| &\leq |\hat{g}_{n,t}(x) - g_n(x)| + |g_n(x) - f(x)| + |f(x) - f_t(x)| \\
&\leq \tilde{\Delta}_{n,t} + d_n + c_t \sigma_{t-1}(x) \\
&\leq \tilde{\Delta}_{n,t} + d_n + c_t,
\end{aligned}
\tag{13}
$$

in which we have made use of Lemma 4, the definition of $d_n$: $d_n = \max_{x \in \mathcal{X}} |f(x) - g_n(x)|$ (Section 2, last paragraph), Equation (12), and the assumption that the posterior variance is upper-bounded by 1. Plugging in the expression of $\tilde{\Delta}_{n,t}$ from Lemma 4 completes the proof. $\square$

**Lemma 6.** *For any filtration $\mathcal{F}_{t-1}$, conditioned on the events $E^f(t)$ and $A_t$, we have that for every $x \in \mathcal{X}$,*

$$\mathbb{P}\left(f_t(x) > f(x)|\mathcal{F}_{t-1}, E^f(t), A_t\right) \geq p,
\tag{14}$$

*in which $p = \frac{1}{4e\sqrt{\pi}}$.*

As shown in the proof of Lemma 8 of [9], the proof of Lemma 6 makes use of the fact that $f_t(x) \sim \mathcal{N}(\mu_{t-1}(x), \beta_t^2 \sigma_{t-1}^2(x))$ since we are conditioning on the event $A_t$, the confidence bound given in Lemma 1 which holds since we are conditioning on the event $E^f(t)$, and the Gaussian anti-concentration lemma. That is, for a Gaussian random variable $X \sim \mathcal{N}(\mu, \sigma^2)$, for any $\beta > 0$, we have that

$$\mathbb{P}\left(\frac{X-\mu}{\sigma} > \beta\right) \geq \frac{\exp(-\beta^2)}{4\sqrt{\pi}\beta}.$$

The next lemma shows that in each iteration $t$, the probability that an unsaturated input is selected can be lower-bounded.

**Lemma 7.** *For any filtration $\mathcal{F}_{t-1}$, conditioned on the event $E^f(t)$, we have that with probability $\geq 1 - \delta/2$,*

$$\mathbb{P}\left(x_t \in \mathcal{X} \setminus S_t|\mathcal{F}_{t-1}\right) \geq P_t,$$

*in which*

$$P_t \triangleq p_t(p - 1/t^2).$$

*Proof.* Note that all probabilities in this proof are conditioned on the event $E^f(t)$ and thus this conditioning is omitted for simplicity. At iteration $t$, the probability that the selected input $x_t$ is unsaturated can be lower-bounded by:

$$\mathbb{P}\left(x_t \in \mathcal{X} \setminus S_t|\mathcal{F}_{t-1}\right) \geq \mathbb{P}\left(x_t \in \mathcal{X} \setminus S_t|\mathcal{F}_{t-1}, A_t\right)\mathbb{P}(A_t) = \mathbb{P}\left(x_t \in \mathcal{X} \setminus S_t|\mathcal{F}_{t-1}, A_t\right)p_t
\tag{15}$$

Next, we attempt to lower-bound $\mathbb{P}\left(x_t \in \mathcal{X} \setminus S_t|\mathcal{F}_{t-1}, A_t\right)$.

Firstly, recall that conditioned on the event $A_t$, $x_t$ is selected by maximizing $f_t$, which is sampled from the GP posterior belief of function $f$. This gives rise to:

$$\mathbb{P}\left(x_t \in \mathcal{X} \setminus S_t | \mathcal{F}_{t-1}, A_t\right) \geq \mathbb{P}\left(f_t(x^*) > f_t(x), \forall x \in S_t | \mathcal{F}_{t-1}, A_t\right). \tag{16}$$

This inequality can be obtained by observing that the event on the right hand side is a subset of the event on the left hand side. Specifically, recall from Definition 1 that $x^*$ is always unsaturated. Therefore, if $f_t(x^*) > f_t(x), \forall x \in S_t$, as a result of the way in which $x_t$ is selected (i.e., $x_t = \arg\max_{x \in \mathcal{X}} f_t(x)$), this guarantees that an unsaturated input will be selected as $x_t$ since at least one unsaturated input ($x^*$) has a larger value of $f_t$ than all saturated inputs.

Next, we assume that both events $E^f(t)$ and $E^{f_t}(t)$ are true, which allows us to derive an upper bound on $f_t(x)$ for all $x \in S_t$:

$$f_t(x) \overset{(a)}{\leq} f(x) + c_t\sigma_{t-1}(x) \overset{(b)}{\leq} f(x) + \Delta(x) = f(x) + f(x^*) - f(x) = f(x^*), \tag{17}$$

in which $(a)$ follows from (12) since here we also assume both events $E^f(t)$ and $E^{f_t}(t)$ are true, and $(b)$ results from the definition of saturated set (Definition 1). Therefore, (17) implies that

$$\mathbb{P}\left(f_t(x^*) > f_t(x), \forall x \in S_t | \mathcal{F}_{t-1}, A_t, E^{f_t}(t)\right) \geq \mathbb{P}\left(f_t(x^*) > f(x^*) | \mathcal{F}_{t-1}, A_t, E^{f_t}(t)\right). \tag{18}$$

Next, we can show that

$$\begin{aligned}
\mathbb{P}\left(x_t \in \mathcal{X} \setminus S_t | \mathcal{F}_{t-1}, A_t\right) &\geq \mathbb{P}\left(f_t(x^*) > f_t(x), \forall x \in S_t | \mathcal{F}_{t-1}, A_t\right) \\
&\overset{(a)}{\geq} \mathbb{P}\left(f_t(x^*) > f(x^*) | \mathcal{F}_{t-1}, A_t\right) - \mathbb{P}\left(\overline{E^{f_t}(t)} | \mathcal{F}_{t-1}\right) \\
&\overset{(b)}{\geq} p - 1/t^2,
\end{aligned} \tag{19}$$

in which $(a)$ follows from some simple probabilistic manipulations and the fact that the event $E^{f_t}(t)$ is $\mathcal{F}_{t-1}$-measurable and thus independent of the event $A_t$, $(b)$ results from Lemma 6 and the fact that the event $E^{f_t}(t)$ holds with probability of at least $1 - 1/t^2$. Combining this inequality with (15) completes the proof.

$\square$

The next lemma presents an upper bound on the expected instantaneous regret of the FTS algorithm.

**Lemma 8.** *For any filtration $\mathcal{F}_{t-1}$, conditioned on the event $E^f(t)$, we have that with probability of $\geq 1 - \delta/2$*

$$\mathbb{E}[r_t | \mathcal{F}_{t-1}] \leq c_t\left(1 + \frac{10}{pp_1}\right)\mathbb{E}\left[\sigma_{t-1}(x_t) | \mathcal{F}_{t-1}\right] + \psi_t + \frac{2B}{t^2},$$

*in which $r_t$ is the instantaneous regret: $r_t = f(x^*) - f(x_t)$, and $\psi_t \triangleq 2(1 - p_t)\sum_{n=1}^N P_N[n]\Delta_{n,t}$.*

*Proof.* To begin with, we define $\overline{x}_t$ as the unsaturated input at iteration $t$ with the smallest (posterior) standard deviation:

$$\overline{x}_t = \arg\min_{x \in \mathcal{X} \setminus S_t} \sigma_{t-1}(x). \tag{20}$$

Following this definition, for any $\mathcal{F}_{t-1}$ such that $E^f(t)$ is true, we have that

$$\begin{aligned}
\mathbb{E}\left[\sigma_{t-1}(x_t) | \mathcal{F}_{t-1}\right] &\geq \mathbb{E}\left[\sigma_{t-1}(x_t) | \mathcal{F}_{t-1}, x_t \in \mathcal{X} \setminus S_t\right] \mathbb{P}\left(x_t \in \mathcal{X} \setminus S_t | \mathcal{F}_{t-1}\right) \\
&\geq \sigma_{t-1}(\overline{x}_t) P_t,
\end{aligned} \tag{21}$$

in which the last inequality follows from the definition of $\overline{x}_t$ and Lemma 7.

Now we condition on both events $E^f(t)$ and $E^{f_t}(t)$, and analyze the instantaneous regret as:

$$\begin{aligned}
r_t = \Delta(x_t) &= f(x^*) - f(\overline{x}_t) + f(\overline{x}_t) - f(x_t) \\
&\overset{(a)}{\leq} \Delta(\overline{x}_t) + f_t(\overline{x}_t) + c_t\sigma_{t-1}(\overline{x}_t) - f_t(x_t) + c_t\sigma_{t-1}(x_t) \\
&\overset{(b)}{\leq} c_t\sigma_{t-1}(\overline{x}_t) + c_t\sigma_{t-1}(\overline{x}_t) + c_t\sigma_{t-1}(x_t) + f_t(\overline{x}_t) - f_t(x_t) \\
&= c_t(2\sigma_{t-1}(\overline{x}_t) + \sigma_{t-1}(x_t)) + \underline{f_t(\overline{x}_t) - f_t(x_t)},
\end{aligned} \tag{22}$$

in which $(a)$ follows from the definition of $\Delta(x)$ and $|f_t(x) - f(x)| \leq c_t\sigma_{t-1}(x)$ for all $x \in \mathcal{X}$ since we assume both events $E^f(t)$ and $E^{f_t}(t)$ are true, and $(b)$ results from the fact that $\overline{x}_t$ is unsaturated. Next, we analyze the expected value of the underlined term given a filtration $\mathcal{F}_{t-1}$:

$$\mathbb{E}\left[f_t(\overline{x}_t) - f_t(x_t)|\mathcal{F}_{t-1}\right]$$

$$= \mathbb{P}(A_t)\mathbb{E}\left[f_t(\overline{x}_t) - f_t(x_t)|\mathcal{F}_{t-1}, A_t\right] + \mathbb{P}(B_t)\sum_{n=1}^{N}P_N[n]\mathbb{E}\left[f_t(\overline{x}_t) - f_t(x_t)|\mathcal{F}_{t-1}, B_{t,n}\right]$$

$$\overset{(a)}{\leq} (1-p_t)\sum_{n=1}^{N}P_N[n]\mathbb{E}\left[f_t(\overline{x}_t) - f_t(x_t)|\mathcal{F}_{t-1}, B_{t,n}\right]$$

$$\overset{(b)}{\leq} (1-p_t)\sum_{n=1}^{N}P_N[n]\mathbb{E}\left[\hat{g}_{n,t}(\overline{x}_t) + \Delta_{n,t} - \hat{g}_{n,t}(x_t) + \Delta_{n,t}|\mathcal{F}_{t-1}, B_{t,n}\right]$$

$$\overset{(c)}{\leq} 2(1-p_t)\sum_{n=1}^{N}P_N[n]\Delta_{n,t} \triangleq \psi_t,$$

(23)

in which $(a)$ follows since when $A_t$ is true, i.e., when $x_t = \arg\max_{x \in \mathcal{X}} f_t(x)$, $f_t(\overline{x}_t) - f_t(x_t) \leq 0$, $(b)$ makes use of Lemma 5 (note that here we are also conditioning on the events $E^f(t)$ and $E^{f_t}(t)$ which is the same as Lemma 5, and that Lemma 5 holds irrespective of the event $B_{t,n}$ since both $E^f_t$ and $E^{f_t}(t)$ are $\mathcal{F}_{t-1}$-measurable) and thus holds with probability of $\geq 1 - \delta/2$, and $(c)$ follows since conditioned on the event $B_{t,n}$ (i.e., $x_t = \arg\max_{x \in \mathcal{X}} \hat{g}_{n,t}(x)$), $\hat{g}_{n,t}(\overline{x}_t) - \hat{g}_{n,t}(x_t) \leq 0$.

Subsequently, we can analyze the expected instantaneous regret by separately considering the two cases in which the event $E^{f_t}(t)$ is true and false respectively:

$$\mathbb{E}\left[r_t|\mathcal{F}_{t-1}\right]$$

$$\leq \mathbb{E}\left[c_t(2\sigma_{t-1}(\overline{x}_t) + \sigma_{t-1}(x_t)) + f_t(\overline{x}_t) - f_t(x_t)|\mathcal{F}_{t-1}\right] + 2B\mathbb{P}\left[\overline{E^{f_t}(t)}|\mathcal{F}_{t-1}\right]$$

$$\leq \mathbb{E}\left[c_t(2\sigma_{t-1}(\overline{x}_t) + \sigma_{t-1}(x_t))|\mathcal{F}_{t-1}\right] + \mathbb{E}\left[f_t(\overline{x}_t) - f_t(x_t)|\mathcal{F}_{t-1}\right] + 2B\mathbb{P}\left[\overline{E^{f_t}(t)}|\mathcal{F}_{t-1}\right]$$

$$\leq \frac{2c_t}{P_t}\mathbb{E}\left[\sigma_{t-1}(x_t)|\mathcal{F}_{t-1}\right] + c_t\mathbb{E}\left[\sigma_{t-1}(x_t)|\mathcal{F}_{t-1}\right] + \psi_t + \frac{2B}{t^2}$$

$$\leq c_t\left(1 + \frac{2}{P_t}\right)\mathbb{E}\left[\sigma_{t-1}(x_t)|\mathcal{F}_{t-1}\right] + \psi_t + \frac{2B}{t^2}.$$

(24)

Note that since $1/(p - 1/t^2) \leq 5/p$ and $p_t \geq p_1$ for all $t \geq 1$,

$$\frac{2}{P_t} = \frac{2}{p_t(p - \frac{1}{t^2})} \leq \frac{10}{pp_t} \leq \frac{10}{pp_1}.$$

(25)

Therefore, (24) can be further analyzed as

$$\mathbb{E}\left[r_t|\mathcal{F}_{t-1}\right] \leq c_t\left(1 + \frac{10}{pp_1}\right)\mathbb{E}\left[\sigma_{t-1}(x_t)|\mathcal{F}_{t-1}\right] + \psi_t + \frac{2B}{t^2},$$

(26)

which completes the proof. $\qquad\square$

Subsequently, we make use of the concentration inequality of super-martingales to derive a bound on the cumulative regret.

**Definition 2.** *Define $Y_0 = 0$, and for all $t = 1, \ldots, T$,*

$$\overline{r}_t = r_t\mathbb{I}\{E^f(t)\},$$

$$X_t = \overline{r}_t - c_t\left(1 + \frac{10}{pp_1}\right)\sigma_{t-1}(x_t) - \psi_t - \frac{2B}{t^2},$$

$$Y_t = \sum_{s=1}^{t}X_s.$$

**Lemma 9.** *Conditioned on Lemma 8 (i.e., with probability of $\geq 1 - \delta/2$), $(Y_t : t = 0, \ldots, T)$ is a super-martingale with respect to the filtration $\mathcal{F}_t$.*

*Proof.*

$$
\begin{aligned}
\mathbb{E}\left[Y_t - Y_{t-1}|\mathcal{F}_{t-1}\right] &= \mathbb{E}\left[X_t|\mathcal{F}_{t-1}\right] \\
&= \mathbb{E}\left[\bar{r}_t - c_t\left(1 + \frac{10}{pp_1}\right)\sigma_{t-1}(x_t) - \psi_t - \frac{2B}{t^2}|\mathcal{F}_{t-1}\right] \\
&= \mathbb{E}\left[\bar{r}_t|\mathcal{F}_{t-1}\right] - \left[c_t\left(1 + \frac{10}{pp_1}\right)\mathbb{E}\left[\sigma_{t-1}(x_t)|\mathcal{F}_{t-1}\right] + \psi_t + \frac{2B}{t^2}\right] \\
&\leq 0,
\end{aligned}
\tag{27}
$$

in which the last inequality follows from Lemma 8 when the event $E^f(t)$ is true; when $E^f(t)$ is false, $\bar{r}_t = 0$ and thus the inequality holds trivially. $\qquad\square$

The Azuma-Hoeffding Inequality presented below will be useful for proving the concentration of the super-martingale $(Y_t : t = 0, \ldots, T)$.

**Lemma 10** (Azuma-Hoeffding Inequality)**.** *Given any $\delta' \in (0, 1)$. If a super-martingale $(Z_T : t = 1, \ldots, T)$, defined with respect to the filtration $\mathcal{F}_t$, satisfies $|Z_t - Z_{t-1}| \leq \alpha_t$ for some constant $\alpha_t$, then for all $t = 1, \ldots, T$ and with probability of at least $1 - \delta'$,*

$$
Z_T - Z_0 \leq \sqrt{2\log(1/\delta')\sum_{t=1}^{T}\alpha_t^2}.
$$

Finally, we can derive an upper bound on the cumulative regret through the following lemma.

**Lemma 11.** *Given $\delta \in (0, 1)$, then with probability of at least $1 - \delta$,*

$$
\begin{aligned}
R_T \leq &c_T\left(1 + \frac{10}{pp_1}\right)\mathcal{O}(\sqrt{T\gamma_T}) + \sum_{t=1}^{T}\psi_t + \frac{B\pi^2}{3} + \\
&\left[c_T\left(1 + \frac{4B}{p} + \frac{10}{pp_1}\right) + \psi_1 + \mathcal{O}(\sqrt{\log T})\right]\sqrt{2T\log\frac{4}{\delta}},
\end{aligned}
$$

*in which $\gamma_T$ is the maximum information gain about $f$ obtained from any set of $T$ observations.*

*Proof.*

$$
\begin{aligned}
|Y_t - Y_{t-1}| = |X_t| &\leq |\bar{r}_t| + c_t\left(1 + \frac{10}{pp_1}\right)\sigma_{t-1}(x_t) + \psi_t + \frac{2B}{t^2} \\
&\overset{(a)}{\leq} 2B + c_t\left(1 + \frac{10}{pp_1}\right) + \psi_t + \frac{2B}{t^2} \\
&\overset{(b)}{\leq} \frac{2Bc_t}{p} + c_t\left(1 + \frac{10}{pp_1}\right) + \psi_t + \frac{2Bc_t}{p} \\
&\leq c_t\left(1 + \frac{4B}{p} + \frac{10}{pp_1}\right) + \psi_t,
\end{aligned}
\tag{28}
$$

in which $(a)$ follows since the posterior variance is upper-bounded by 1, $(b)$ follows since $2B \leq 2Bc_t/p$ and $2B/t^2 \leq 2Bc_t/p$.

This allows us to apply the Azuma-Hoeffding Inequality (Lemma 10) by using an error probability of $\delta/4$,

$$
\begin{aligned}
\sum_{t=1}^{T} \bar{r}_t &\le \sum_{t=1}^{T} c_t \left(1 + \frac{10}{pp_1}\right) \sigma_{t-1}(x_t) + \sum_{t=1}^{T} \psi_t + \sum_{t=1}^{T} \frac{2B}{t^2} + \\
&\quad \sqrt{2 \log \frac{4}{\delta} \sum_{t=1}^{T} \left[c_t \left(1 + \frac{4B}{p} + \frac{10}{pp_1}\right) + \psi_t\right]^2} \\
&\le c_T \left(1 + \frac{10}{pp_1}\right) \sum_{t=1}^{T} \sigma_{t-1}(x_t) + \sum_{t=1}^{T} \psi_t + \frac{B\pi^2}{3} + \\
&\quad \left[c_T \left(1 + \frac{4B}{p} + \frac{10}{pp_1}\right) + \psi_1 + \mathcal{O}(\sqrt{\log T})\right] \sqrt{2T \log \frac{4}{\delta}} \\
&= c_T \left(1 + \frac{10}{pp_1}\right) \mathcal{O}(\sqrt{T\gamma_T}) + \sum_{t=1}^{T} \psi_t + \frac{B\pi^2}{3} + \\
&\quad \left[c_T \left(1 + \frac{4B}{p} + \frac{10}{pp_1}\right) + \psi_1 + \mathcal{O}(\sqrt{\log T})\right] \sqrt{2T \log \frac{4}{\delta}},
\end{aligned}
\tag{29}
$$

which holds with probability $\ge 1 - \delta/4$. The last inequality follows since $c_t$ is increasing in $t$, $\sum_{t=1}^{T} 1/t^2 = \pi^2/6$, and $\psi_t \le \psi_1 + \mathcal{O}(\sqrt{\log t})$ for all $t \in \mathbb{Z}^+$ which is ensured by the way in which we choose the sequence $p_t$, i.e., such that $(1 - p_t)c_t \le (1 - p_1)c_1$ for all $t \in \mathbb{Z}^+ \setminus \{1\}$. Lastly, note that the event $E^f(t)$ holds with probability $\ge 1 - \delta/4$, i.e., $\bar{r}_t = r_t$ with probability $\ge 1 - \delta/4$. In the last equality, we made use of the fact that $\sum_{t=1}^{T} \sigma_{t-1}(x_t) = \mathcal{O}(\sqrt{T\gamma_T})$ which is proved by Lemmas 5.3 and 5.4 of [52]. Taking into account the error probability of Lemma 8 ($\delta/2$), which is required for $(Y_t : t = 0, \dots, T)$ to form a super-martingale, completes the proof. $\qquad \square$

Finally, we are ready to prove Theorem 1. Recall that $c_t = \mathcal{O}\left(\left(B + \sqrt{\gamma_t + \log(1/\delta)}\right)\sqrt{\log t}\right)$. Therefore,

$$
\begin{aligned}
R_T &= \mathcal{O}\left(\frac{1}{p_1}\left(B + \sqrt{\gamma_T + \log\frac{1}{\delta}}\right)\sqrt{\log T}\sqrt{T\gamma_T} + \sum_{t=1}^{T} \psi_t + \right. \\
&\quad \left. \left(B + \frac{1}{p_1}\right)\left(B + \sqrt{\gamma_T + \log\frac{1}{\delta}}\right)\sqrt{\log T}\sqrt{T\log\frac{1}{\delta}}\right) \\
&= \mathcal{O}\left(\left(B + \frac{1}{p_1}\right)\sqrt{T\log T\gamma_T \log\frac{1}{\delta}\left(\gamma_T + \log\frac{1}{\delta}\right)} + \sum_{t=1}^{T} \psi_t\right) \\
&= \tilde{\mathcal{O}}\left(\left(B + \frac{1}{p_1}\right)\gamma_T\sqrt{T} + \sum_{t=1}^{T} \psi_t\right).
\end{aligned}
\tag{30}
$$

## D Further Experimental Details and Results

All experiments reported in this work are run on a computer with 48 cores of Xeon Silver 4116 (2.1Ghz) processors, RAM of 256GB, and 1 NVIDIA Tesla T4 GPU. For fair comparisons, in all experiments, the same random initializations are used by all methods.

### D.1 Optimization of Synthetic Functions

In the synthetic experiments, we sample the objective functions from a GP with a length scale of 0.03. The functions are defined on a 1-dimensional discrete domain uniformly distributed in $[0, 1]$,

Figure 4: Performances in the most general setting in which $t_n$ is increasing (green curve) for the (a) landmine detection, (b) Google glasses and (c) mobile phone sensors experiments. The specific experimental setup is described in Appendix D.2.1. The results correspond to Fig. 3 in the main paper.

with size $|\mathcal{X}| = 1,000$. The output values of all functions $f(x), \forall x \in \mathcal{X}$ are normalized into the range of $[0,1]$. Whenever an input $x$ is queried, the corresponding noisy observation is obtained by adding a zero-mean Gaussian noise $\mathcal{N}(0, \sigma^2)$ where $\sigma^2 = 0.01$ to the corresponding function value $f(x)$ (Section 2, first paragraph). For a sampled objective function for the target agent, we generate the objective functions of the other agents, as well as their observations, in the following way. For agent $\mathcal{A}_n$, we go through every input in the entire discrete domain, and for each input, we add either $d_n$ or $-d_n$ to the corresponding output function value with probability $0.5$ each, after which the resulting value is used as the objective function value of the agent $\mathcal{A}_n$. This step ensures the validity of the definition of $d_n$ as the maximum difference between the objective function of the target agent $\mathcal{A}$ and that of agent $\mathcal{A}_n$: $d_n = \max_{x \in \mathcal{X}} |f(x) - g_n(x)|$ (Section 2, last paragraph). Next, we randomly sample $t_n$ inputs from the entire discrete domain, and for each sampled input, we obtain a noisy output observation by adding a zero-mean Gaussian noise: $\mathcal{N}(0, \sigma^2)$ where $\sigma^2 = 0.01$, to the corresponding function value. Subsequently, following the procedures described in the first paragraph of Section 3.1, every agent $\mathcal{A}_n$ applies RFF approximation using its own $t_n$ observations (input-output pairs) to derive the RFF approximation parameters $\nu_n$ and $\Sigma_n$ and hence to draw a sample of $\omega_n$, which is the parameter to be passed to and used by the target agent $\mathcal{A}$. Finally, after receiving the parameters $\omega_n$'s from all other agents, the target agent starts to run the FTS algorithm (Algorithm 1).

## D.2 Real-world Experiments

### D.2.1 Results in the Most General Setting with Increasing $t_n$

Here we perform additional experiments in the most general setting of our FTS algorithm (Section 3.1, last paragraph): (a) information can be received from every agent $\mathcal{A}_n$ before every iteration instead of only before the first iteration, and (b) every $\mathcal{A}_n$ may also be performing black-box optimization tasks (possibly also using FTS), such that $\mathcal{A}_n$ may collect more observations (i.e., $t_n$ may increase) between different rounds of communication. We use the three real-world experiments (Section 5.2) to investigate the performances in this setting, and compare the performances with those in the simpler setting where communication is allowed only before the first iteration.

Here we describe the detailed experimental setup for the experiments in this section. Before the first iteration of the FTS algorithm, every agent $\mathcal{A}_n$ for $n = 1, \ldots, N$, who has completed $t_n = 50$ iterations of its own BO task (we use standard TS here for simplicity, but it can be replaced by FTS in which $\mathcal{A}_n$ is the target agent), passes the first message to the target agent $\mathcal{A}$. Next, before every iteration $t > 1$ of the FTS algorithm (Algorithm 1), every agent $\mathcal{A}_n$ runs *one more iteration* of its own BO task, calculates the updated RFF approximation parameters $\nu_{n,t}$ and $\Sigma_{n,t}$ (3), samples a new $\omega_{n,t}$ from its posterior belief: $\omega_{n,t} \sim \mathcal{N}(\nu_{n,t}, \sigma^2 \Sigma_{n,t}^{-1})$, and finally passes the sampled $\omega_{n,t}$ to the target agent $\mathcal{A}$. Then, $\mathcal{A}$ uses the received updated information to run iteration $t$ of the FTS algorithm. After this, the information from every agent $\mathcal{A}_n$ is updated and sent to $\mathcal{A}$ again, and the FTS algorithm proceeds to the next iteration $t + 1$. As a result, every $t_n, \forall n = 1, \ldots, N$ increases by 1 after every iteration of FTS.

The performances in all three experiments are shown in Fig. 4, in which FTS outperforms standard TS in both settings for all experiments. The figure also shows that in the most general setting in which $t_n$ is increasing, the performances for the two activity recognition experiments (Google glasses and mobile phone sensors experiments) are improved, whereas the performances for the landmine detection experiment are comparable in both settings. Note that the most general setting with increasing $t_n$ may not necessarily lead to better performances: Although using more observations from those agents with similar objective functions to the target agent can give more useful information and hence potentially benefit the FTS algorithm, more observations from *heterogeneous agents* may turn out to hurt the performance of FTS since the information from these agents are actually harmful for the BO task of the target agent.

### D.2.2 More Experimental Details

In all real-world experiments, we use length scale $= 0.01$ to generate the random features (Appendix A) and $\sigma^2 = 10^{-6}$ in the RFF approximation using equations (2) and (3).

**Landmine Detection.** This dataset, downloadable from `http://www.ee.duke.edu/~lcarin/LandmineData.zip`, consists of the data from 29 landmine fields, with each field associated with a dataset for landmine detection. The dataset of each landmine field is made up of a number of input-output pairs, each corresponding to a location; for every location, the input includes 9 features extracted from radar images and the output is a binary label indicating whether the location contains landmines. The number of data points (input-output pairs) of every field ranges from 445 to 690, with a mean of 511; for every field, we use 50% of the data points as the training set, and the other 50% as the validation set. We use support vector machines (SVM) as the predictive model, and tune two SVM hyperparameters: RBF kernel parameter in the range of $[0.01, 10]$, and L2 regularization parameter in $[10^{-4}, 10]$. For every queried hyperparamter setting, the SVM model is trained on the training set using this particular set of hyperparameters, and evaluated using the validation set to produce the reported performances. As mentioned in the main text, the dataset of the landmine fields are significantly imbalanced, i.e., there are considerably more locations without than with landmines. Specifically, the percentage of positive samples (i.e., locations with landmines) in different landmine fields ranges from 2.9% to 9.4%, with a mean of 6.2%. Therefore, for this dataset, validation error is inappropriate since an all-zero prediction would result in very low classification error. Hence, we use the Area Under the Receiver Operating Characteristic Curve (AUC) which is a more appropriate metric when evaluating the performance of ML models on imbalanced datasets.

**Activity Recognition Using Google Glasses.** This dataset consists of two-hour long sensor data collected using Google glasses from 38 participants, while the participants are performing different activities such as eating. The dataset can be downloaded from `http://www.skleinberg.org/data/GLEAM.tar.gz`. For every participant, we group the sensor data into different time windows; for each time window, we calculate the statistics (i.e., mean, variance and kurtosis) of different sensor measurements within this time window, and use them as the features (57 features in total are extracted from each time window); the label for each time window is a binary value indicating whether the participant is eating or conducting other activities during this time window. As a result, for every participant, each time window produces a data point, i.e., an input-output pair. The number of data points for every participant ranges from 242 to 3416 with an average of 1930. For every participant, we randomly select 100 data points as the validation set, and use the remaining data points as the training set. We use logistic regression (LR) as the activity prediction model for every participant, and tune 3 hyperparameters of LR: the batch size in the range of $[20, 60]$, the L2 regularization parameter in $[10^{-6}, 1]$, and the learning rate in $[0.01, 0.1]$. Following the common practice for using LR and neural network models, the inputs are pre-processed by removing the mean and dividing by the standard deviation.

**Activity Recognition Using Mobile Phone Sensors.** This dataset, which can be downloaded from `https://archive.ics.uci.edu/ml/datasets/Human+Activity+Recognition+Using+Smartphones`, contains measurements from mobile phone sensors (accelerometer and gyroscope) involving 30 subjects. 561 features were provided together with the dataset, with each set of features associated with a corresponding label indicating which one of the six activities the subject is performing. Therefore, the activity recognition problem for every subject corresponds to a 6-class classification problem. The number of data points (input-output pairs) possessed by the subjects ranges from 281 to 409 with a mean of 343. For every subject, we use 50% of the data points as the training set, and the remaining 50% as the validation set. We again use LR as the activity recognition

Figure 5: Additional results for the other $5$ target agents for the landmine detection experiment ($M = 100$).

model, and the tuned hyperparameters, as well as their ranges, are the same as those in the activity recognition experiment using Google glasses.

### D.2.3 Additional Results for More Agents

In this section, we present additional experimental results for the three real-world experiments (Section 5.2). Note that as mentioned in Section 5.2 (last paragraph), the results presented in Fig. 3 in the main text correspond to using the first agent (of the 6 agents used to produce the results in Fig. 2 in the main text) as the target agent for every experiment. Meanwhile, the additional results shown in this section (Figs. 5, 6 and 7) correspond to using each of the remaining 5 agents (agents 2 to 6) as the target agent. Note that since all three real-world datasets contain heterogeneous agents (Section 5.2, first paragraph), it is unreasonable to expect FTS to always outperform standard TS for all agents. Instead, as shown in the figures, FTS performs better than TS for some agents, and comparably with TS for other agents.

Figure 6: Additional results for the other 5 target agents for the activity recognition experiment using Google glasses ($M = 100$).

Figure 7: Additional results for the other 5 target agents for the activity recognition experiment using mobile phone sensors ($M = 100$).