[Reviews · NeurIPS 2020]

Review 1

Summary and Contributions: * The paper studies collaborative Bayesian Optimization in a Federated Learning setting. * The core idea is to utilize random Fourier feature approximations of GPs to avoid exchanging raw observations * The paper derives regret bounds for the approach, and compares its empirical performance to that of some (not really appropriate) alternatives in simulations.

Strengths: * This paper provides an interesting view of Bayesian Optimization in the context of Federated Learning, and seems practically relevant (though I note my limited exposure in this domain). * Using RFF GPs is an elegant way to ensure data is not shared between agents * The theoretical contribution of the regret bound appears sound, albeit not particularly surprising (see below).

Weaknesses: * The sublinear asymptotic regret bound is not particularly interesting, as it hinges on the sequence p_t converging to 1, meaning that information contributed by other agents will asymptotically be washed out by purely the agent's own information. So without the specifics of the bound, this is very intuitive, and since BO is a global optimization methodology there isn't concern that the agent will get stuck in some minimum that would prevent it from recovering from being mislead early on. However, the specific form of the bound does provide some insight into the dependency on the number of features. * The comparison with RGPE and TAF do not really seem appropriate, as those methods were never developed with the setting considered in this paper in mind. If these are the only points of comparison it's fine to have them in the paper, but the authors should state clearly that the provided comparisons are really against non-competing methods. * The authors only consider RFF, wheres it seems that a number of other models (stochastic variational GPs, inducing point methods) could be used as alternative ways of avoiding to share raw observations between agents.

Correctness: * The theoretical contribution of the regret bound appears sound * The empirical results are presented well, though further details on the simulation environment would be helpful to better understand the results. * The authors state that "when x_t is selected using \omega_n from an agent, FTS only needs to solve a linear optimization problem of dimension M" - This seems wrong to me, as the cosine non-linearities in the FFs would imply that this is a (hard) nonlinear optimization problem.

Clarity: * The paper is quite well written, and is relatively easy to follow. * I feel that the authors should provide more intuition for why FTS performs better than TS. Essentially, my understanding here is that each agent benefits (albeit implicitly) from having (N+1) times as many observations. This should be discussed in the paper. Also, the results will depend heavily on simulation environment, which should be discussed in more detail.

Relation to Prior Work: Yes

Reproducibility: Yes

Additional Feedback: - the authors state that "the Gaussian process (GP) model, which is the most commonly used surrogate model in BO since it is required for deriving the theoretical convergence guarantee". This statement doesn't really make sense - theoretical convergence guarantees and practical performance are very different things. - regarding the statement of the update of the computational cost: while optimizing given \omega_n is cheap, sampling \omega_n for the other agent is not (requires drawing a sample from a MVN gaussian posterior, which essentially amounts to computing a root decomposition of that agent's Sigma_t^{-1}, which grows linearly in t) - while the authors mention privacy-preserving sharing of data (e.g. differential privacy), I feel the paper could benefit by further exploring these thoughts in some more detail. It seems like the data in this setting is small enough that that may well be feasible (compared to DNN-based FL approaches) - what is the horizontal line in Fig 2? - the landmine example seems very out of place: Why would anyone want to run a FL algorithm for landmine detection that avoids sharing the classifiers between landmine fields? - the paper makes a number of hyperbolic statements, that I don't think are appropriate for a research paper: "immense potential", "considerably superior" ********** Post-Rebuttal Comments ************** The authors' response did clarified a couple of the points I raised / misunderstandings I had. I am not particularly confident that using DIRECT for optimizing line 6 is the best choice, especially for higher-dimensional problems. I don't really feel swayed either way by the response.


Review 2

Summary and Contributions: This paper presents a Bayesian Optimization approach extending Thompson sampling to cases where several proxies of the objective function co-exist, and lead to as many Gaussian Process models relying on observations from the different proxies (one of them being the objective function itself). The main idea is to leverage this multiplicity while keeping respective observations private to the individual models, namely by exchanging information about GP parameters. Things are made particularly simple here by adopting shared random features, so that one ends up with Bayesian (Gaussian) linear models, and the information exchange amounts to communicating posterior distributions of model parameters, via posterior means and covariances in the considered Gaussian case. Then the modification of Thompson sampling is plainly done by sampling from the distribution of optimizers of one of the considered posterior GP models, either from the one associated with the objective function, or with some probability from one of the GP models associated with the proxies. Here the probability of choosing the first case increases to 1 at a prescribed pace, and the conditional probabilities of the choosing each of remaining proxy GPs otherwise are prescribed, typically uniform among the remaining models (or « agents ») at hand. A theoretical result is established in terms of cumulative regret, that guarantees sublinear growth (and quantifies it in more detail) with arbitrary high probability given an adequate scheme for the probability of choosing the agent associated with the actual objective function. Furthermore, it is shown on GP realizations and on several real-world experiments that i) the proposed « federated Thompson Sampling » may achieve fast decrease of the simple regret than standard Thompson Sampling and ii) can provide substantial wall-clock time savings.

Strengths: The idea of federating various agents in order to help and speed-up Bayesian Optimization is great and of huge potential usefulness in machine learning and in a variety of application fields. The approach introduced here is building up on Thompson sampling, one of the canonical Bayesian Optimization algorithms, and comes at the same time with theoretical guarantees (NB : I did not have a chance to thoroughly check the proof) and promising speed-ups observed in numerical experiments.

Weaknesses: It feels like the proposed approach could be cast as some specific instance of Thompson sampling scheme under a model mixture, and that by highlighting that one should probably in turn explore further the literature on BO under model mixtures. Also, it appears that the mixing scheme used here is quite simplistic and not accounting for the respective levels of fidelity of the different agents, offering considerable room for improvement (although incorporating this in the theoretical result might be quite involved). Besides this, while multi-fidelity / multi-source / multi-model BO has grown in the last years, there is not much account of that in the paper and it might well be relevant to better situate the contribution within these frameworks. Last but not least, I liked very much the idea of appealing to a common realization of random features, but I am also wondering to what extent this could be applied to more generic hyperparameters (e.g. kernel hyperparameters beyond coefficients in the linear case).

Correctness: I was a bit puzzled in lines 106-108 by the random kernel approximation followed by a so-called theoretical guarantee where I don’t really see how the randomness is accounted for. Also, in lines 131-132, the authors claim that “This further suggests that the absolute function values are upper bounded”; yet for this one requires a bounded kernel. Have according assumptions been made (e.g. compact domain and continuous kernel)? As for the correctness of the theoretical proposition, I did not have a chance to thoroughly check the proof. Further minor comments/questions follow: 1. “M \geq 1-dimensional random features“ in line 105 is kind of confusing 2. In line 114, in “which contain M2 and M parameters respectively”, employing “parameters” may be misleading 3. In line 299, “We treat one of the participants”, is the one fixed or random? More detail would help. In line 313, “Fig. 2 shows the (averaged) best performance”, could you explain in more detail why the focus is solely on the “best”? Is it simply the sense of taking the best observed point/response of a sequence as outcome?

Clarity: While I found the paper very well written overall, I have the feeling that I struggled a bit more than I should have trying to understand how the method works. Perhaps there is a way to formulate things in a simple manner, revolving around the idea that x^* is drawn at random from a mixture.

Relation to Prior Work: In relation to the previous points, I expect that once the proposed approach is seen as Thompson sampling under a mixture of models, a door is opened to the literature on related topics. I would be keen on hearing from the authors how they position their work with respect to this literature, that of course requires some additional bibliographical work. In a different flavour, while it is clear that the present approach offers so nice specificities related to information privacy, it still appears related to a stream of works around multi-fidelity BO, multi-source BO, multi-model BO, etc. and I think that an additional effort to situate the paper within this stream would potentially be worthwhile.

Reproducibility: Yes

Additional Feedback: Regarding privacy, it might be relevant to discuss to what extent knowing parameters of one or ther other agent may inform or not about them (i.e., even if the input/output tuples are unknown, one still gets some info). Added post-rebuttal: To me the rebuttal is fair.


Review 3

Summary and Contributions: This paper proposed a framework of federated Bayesian optimization for distributed collaborative zero-order optimization via Thompson sampling with approximation. Random Fourier features (RFF) approximation is adopted for GP to reduce the complexity. Theoretical analysis shows the convergence of FBO. Experiments with synthetic functions and real datasets validate the claims of faster convergence.

Strengths: * Novel framework of federated Bayesian optimization. * Theoretical analysis proves the convergence * Experiments validate the claims.

Weaknesses: * It seems quite computation- and communication- intensive and may limit its applicability in the real-world setting. * The idea is quite a natural extension into federated learning setting.

Correctness: Yes.

Clarity: Yes.

Relation to Prior Work: Yes.

Reproducibility: No

Additional Feedback:


Review 4

Summary and Contributions: The authors present an approach to perform Bayesian Optimization in a federated manner. The method applies in particular in settings, where no gradient information can be shared between agents, such as gradient-free hyper-parameter optimization for deep networks. Instead of passing direct information about a surrogate model (modeled as a Gaussian Process) between agents, the authors propose to instead pass random Fourier features of said GPs. To enable efficiency of communication between agents, federated Thompson sampling and random Fourier features are proposed as an effective solution. Finally, the authors provide a regret bound analysis applicable for agents with significantly different objective functions.

Strengths: The author's contribution is timely, relevant and within an area that is currently receiving significant attention. The main regret analysis theorem will be of interest to the community. The work is evaluated both on real and synthetic datasets.

Weaknesses: Approach-wise, the work is a combination and modification of three established techniques, i.e. mainly Bayesian Optimization with GPs, Random Fourier Features and a distributed Thompson Sampling approach. Experiments: The authors validate the convergence and data efficiency first on a simple synthetic GP model. Since GP generated data is likely easiest for a method based on GPs, another experiment with increasingly challenging synthetic optimization functions would have been of interest. Previous concerns highlighted in the first version of this review have been addressed by the authors during the rebuttal phase and this review has been updated accordingly.

Correctness: The overall methodology appears correct, but the reviewer did not have time to verify all the details of the proof of the main theorem in the appendix.

Clarity: The paper is well written and the method is clearly described.

Relation to Prior Work: The authors have addressed feedback comments on additional related work in their rebuttal.

Reproducibility: Yes

Additional Feedback: The reviewer thanks the authors for their modifications to the paper following the initial review.

[Author Response · NeurIPS 2020]

We would like to thank all reviewers for your valuable feedback to help improve our paper.

**Reviewer #1:** 3. Weaknesses: • RGPE and TAF are the only points of comparison that we are aware of, and they can be easily adapted for the federated setting (lines 174-177). We'll revise the paper to make it clearer that RGPE and TAF are not developed for the setting considered in this paper. • We agree that other approximation techniques for GP (e.g., inducing point methods) could also be used to avoid the sharing of raw data. It would be interesting (and potentially challenging) to explore whether they can also tackle the key challenges of federated BO in a principled way. 4. Correctness: • You are right that selecting $x_t$ using $\omega_n$ is in fact a non-linear optimization problem due to cosine non-linearities. Fortunately, it can still be efficiently solved in our experiments using the DIRECT method (taking on average $0.76$s/iteration for $M = 200$, landmine experiment). Thank you for pointing this out and we'll correct this. 8. Additional feedback: • Regarding the computational cost of sampling $\omega_n$, we'd like to clarify that the size of the matrix $\Sigma_t^{-1}$ is in fact $M \times M$ (Eq. 3) and independent of $t$, where $M$ is the number of random features and hence a constant. For the values of $M$ used in our experiments (up to $200$), we've found the computational cost to be reasonable, i.e., in the order of $0.1$s. We'll discuss this in the paper. • The horizontal line in Fig. 2 represents the performance of TS.

Thank you for your careful review and we'll also address all your other comments when revising our paper, e.g., discussing why FTS performs better than TS and revising inappropriate statements.

**Reviewer #2:** 3. Weaknesses: • Your suggestion on perceiving our algorithm as BO under model mixtures is very interesting, and we'll explore whether it can be cast in this alternative interpretation. We'll explore these literature, and also add references to multi-fidelity/multi-source BO as you suggested. • We agree that considering the level of fidelity of different agents (i.e., similarity to the target agent) is an interesting extension and have discussed it in lines 344-346. 4. Correctness: • The theoretical guarantee in lines 106-108 is in fact a high-probability guarantee, which we omitted for simplicity. We'll add this after revision. • For the claim in lines 131-132, you are right that we have assumed that the kernel is bounded (line 97). We'll revise and make our assumptions clearer.

Thank you for your constructive suggestions. We'll also take into account all your other comments to revise our paper.

**Reviewer #3:** 3. Weaknesses: • We would like to clarify that we have shown that our FTS algorithm is efficient in terms of both computation (please refer to Fig. 3) and communication (please refer to Fig. 2).

**Reviewer #4:** 3. Weaknesses: • *Experiments*: It's indeed of interest to us to optimize challenging functions, which we have done using the real-world experiments. Meanwhile, we use synthetic experiments to verify the practical relevance of theoretical results and to investigate the behavior of our method, which is made simpler by easier synthetic functions. Thompson sampling (TS) in the non-federated setting simply runs standard TS for a single agent, with no communication with any other agent. In Figs. 2 and 3, the number of sampling iterations for TS is $50$, which is the same for all methods. The performance advantage of FTS is consistent across all 3 real-world experiments, showing that it's stable against variations in various factors such as optimization function, number of agents, etc. We'll add more details on vanilla TS and more discussions about the experiments.

• *Secondly*, as you suggested, we added an experiment to investigate scalability w.r.t. the number of agents (refer to the figure, landmine experiment). The results show that our FTS is more scalable w.r.t. the number of agents, which verifies our analysis in lines 194-202. Regarding our limitations, Fig. 1c in fact contains a failure case. It shows that when all other agents are heterogeneous, FTS can converge slightly slower than TS if $p_t$ doesn't grow sufficiently fast. However, Fig. 1c also shows that FTS still converges to the same final performance as TS, and this limitation can be alleviated by making $p_t$ grow faster (red curve). Another limitation on

susceptibility to advanced privacy attacks and opportunity for future work are discussed in the Broader Impact section. • *Privacy Preservation (line 50)*: You are right that RFF only allows us to *retain the raw data* and avoid transmitting it, so that we can pass RFF parameters *in the same way as standard federated learning* (lines 60-63). We'll revise the paper to reflect this. We'll also explore effective means to preserve privacy in future work (Broader Impact section). • *Communication Efficiency (line 189)*: We agree that we can only claim our FTS is more efficient in communication for a fixed experimental setting and a fixed approximation quality (fixed $M$). We have shown this in our experiments (Fig. 2). We'll revise to make our claim more specific and accurate. 6. Relation to prior work: • We agree that previous works on parallel/distributed BO/TS are also related. We'll cite these works such as the one you suggested, and discuss their relationship with our work. Please also note that there are fundamental differences: they usually optimize a single objective function while we need to consider different functions from (potentially highly) heterogeneous agents; they usually allow the sharing of raw data. 8. Additional feedback: • We would like to clarify that sharing of random features $\phi(x)$ can be achieved by sharing a *finite* number of parameters whose dimensions depend on $D$ and $M$ (Appendix A), even if $\mathcal{X}$ is uncountable.

Thank you very much for you thorough and constructive review, and we'll also address your other comments to revise our paper. We hope that our clarifications and additional results can improve your opinion of our work.

[Meta-Review · NeurIPS 2020]

The paper contains a range of interesting ideas. In its current (submitted form), it's not a great paper, but OK. However, we expect the authors make the changes the reviewers asked for (and the authors committed to) to make raise the paper to the next level. Specifically: 1. Adapt RGPE and TAF to the federated setting (that is 'easily possible') and provide a comparison. 2. Discuss whether/how alternative GP approximations (e.g., inducing point methods) can be used in this setting (i.e., avoid to share raw observations between agents). 3. Interpret the current approach as a specific instance of Thompson sampling scheme under a model mixture and relate to existing literature on BO under model mixtures. 4. Discuss a setting where agents have different levels of fidelity (if possible) and discuss your approach in the context of existing literature on multi-fidelity / multi-source / multi-model BO. 5. Discuss to what extent the proposed approach could be applied to more generic hyperparameters (e.g. kernel hyperparameters beyond coefficients in the linear case). 6. Place the paper in the context of previous works on parallel/distributed BO/TS.